# Self-utility distance as a computational approach to understanding self-concept clarity

Josué García-Arch [1,2] ✉, Christoph W. Korn [3] & Lluís Fuentemilla [1,2,4]

Self-concept stability and cohesion are crucial for psychological functioning and well-being, yet the mechanisms that underpin this fundamental aspect of human cognition remain underexplored. Integrating insights from cognitive and personality psychology with reinforcement learning, we introduce Self-Utility Distance (SUD)—a metric quantifying the dissimilarities between individuals' self-concept attributes and their expected utility value. In Study 1 ($n = 155$), participants provided self- and expected utility ratings using a set of predefined adjectives. SUD showed a significant negative relationship with Self-Concept Clarity that persisted after accounting for individuals' Self-Esteem. In Study 2 ($n = 323$), we found that SUD provides incremental predictive accuracy over Ideal-Self and Ought-Self discrepancies in the prediction of Self-Concept Clarity. In Study 3 ($n = 85$), we investigated the mechanistic principles underlying Self-Utility Distance. Participants conducted a social learning task where they learned about trait utilities from a reference group. We formalized different computational models to investigate the strategies individuals use to adjust trait utility estimates in response to environmental feedback. Through Hierarchical Bayesian Inference, we found evidence that participants utilized their self-concept to modulate trait utility learning, effectively avoiding the maximization of Self-Utility Distance. Our findings provide insights into self-concept dynamics that might help understand the maintenance of adaptive and maladaptive traits.

Establishing a clear, stable, and cohesive understanding of who we are is fundamental for navigating the complexities of daily life. This instrumental aspect of human cognition is captured by the construct of Self-Concept Clarity, typically defined as the extent to which self-concepts are clearly and confidently defined, internally consistent, and stable over time[1,2]. The predictive power of Self-Concept Clarity is pervasive across diverse domains. For example, higher Self-Concept Clarity has been associated with well-being and psychological adjustment[3–5], relationship quality[6], problem-solving during social conflict[7], educational achievement[8], occupational success[9], reduced job burnout[10], and better mental health[1,2,11–14]. Despite its prominent role in psychological research, much of the existing literature has predominantly focused on the outcomes associated with Self-Concept Clarity, leaving its underlying mechanisms underexplored. Here, we aim to provide a fresh perspective that could help understand the underpinnings of Self-Concept Clarity, rooted in individuals' perceived adaptation to their current life situations.

Considering the importance of Self-Concept Clarity for understanding different psychological processes, it is essential to build a thorough understanding of the potential mechanisms that drive its formation and maintenance. However, this goal still remains elusive (but see refs. 15–18). For example, the definition of Self-Concept Clarity incorporates notions of certainty, temporal stability, and internal consistency of self-related attributes[1,2]. Yet, research indicates that measures based on these indicators do not accurately predict global indicators of Self-Concept Clarity, nor are they strongly intercorrelated[12]. Moreover, investigations into the potential mechanisms underlying Self-Concept Clarity have primarily focused on its relationship with broad adjacent constructs like self-esteem, yielding mixed findings regarding their causal directionality, or mutual influence[19–22]. Therefore, research might benefit from incorporating narrower, mechanistically informed predictors to shed light on the dynamics underlying Self-Concept Clarity. This approach would not only deepen our understanding of Self-Concept Clarity but also potentially clarify its relationship with other psychological constructs and inform interventions capable of improving it[23].

[1]Department of Cognition, Development and Education Psychology, Faculty of Psychology, University of Barcelona, Barcelona, Spain. [2]Institute of Neuroscience (UBNeuro), University of Barcelona, Barcelona, Spain. [3]Section Social Neuroscience, Department of General Psychiatry, University of Heidelberg, Heidelberg, Germany. [4]Bellvitge Institute for Biomedical Research, 08908 Hospitalet de Llobregat, Barcelona, Spain. ✉e-mail: j.garcia.arch@ub.edu

Drawing on insights from research in self-concept dynamics, personality psychology, and reinforcement learning, we introduce 'Self-Utility Distance' as a predictor and potential mechanism related to Self-Concept Clarity. Self-utility distance encapsulates the dissimilarity between individuals' current self-concept—comprising various personality traits such as 'Sociable' or 'Anxious'—and their subjective estimations of how these personal characteristics contribute to maximizing the rewards or avoiding the negative outcomes available in their current environments. Put simply, Self-Utility Distance reflects the distance between current self-attributes and their 'expected utility values'. As an example of high Self-Utility Distance, consider an individual who identifies strongly as being "independent" (e.g., tends to be self-reliant, tends to work autonomously, likes to do plans by herself). If this person is part of a work culture that heavily emphasizes teamwork and collaborative processes (e.g., frequent team meetings, shared projects, group work), they might perceive a low utility in their independence, seeing it as less conducive to gaining rewards (e.g., team-based bonuses, promotions) or avoiding negative outcomes (e.g., job loss). Conversely (low Self-Utility Distance), If the individual works in a setting that values autonomous work (e.g., remote work, flexible project choices), their independent nature would align closely with the environment's demands. In this context, the individual might perceive high utility in their independence, as it might enhance their ability to achieve rewards (e.g., recognition for individual contributions, opportunities for self-directed projects) and prevent negative outcomes (e.g., conflicts over team roles). Although its definition might vary slightly across fields, *utility* is a measure reflecting the expected cumulative rewards associated with a particular state, decision, or action[24,25]. Reinforcement learning (RL) models capitalize on the notion that utility computations guide individuals' adaptation strategies by helping them select sets of behaviors (referred to as *policies* in RL) that maximize long-term cumulative rewards. Through interactions with their environment, individuals learn to update their utility estimates and optimize their behavior accordingly, bolstering adaptation. For example, an RL agent might learn that a more competitive strategy yields higher rewards than a cooperative one and progressively adjust its policy[26]. In this context, difficulties in meeting environmental demands are typically seen as resulting from inadequate learning or holding an inaccurate or incomplete model of the environment. These principles share some parallels with other theories strongly focused on how biological agents update their models of the world to promote adaptation[27]. Despite the theoretical and mathematical elegance of RL models and their successful application for explaining a multitude of phenomena, they remain limited to explaining human adaptation to real-life situations.

Unlike RL agents, whose behavior is tightly governed by environmental feedback, humans exhibit self-representations and behavioral tendencies that do not adjust as flexibly as actions and policies in RL. For example, there is evidence that although most people would like to modify some aspect of their personal attributes[28], having a clear intention for personal change does not necessarily lead to actual change[29]. Moreover, evidence from different fields indicates a general tendency towards stability in our self-views and behavioral tendencies. Different models from personality research suggest that the time course of our behavior has fluctuations. However, there is a prevailing tendency for our actions, thoughts, and emotions to reliably return to characteristic baseline patterns[30–32]. Importantly, this stability cannot be attributable to environmental consistency[33,34]. In line with this notion, research from cognitive and experimental psychology suggests that our self-concept is governed by the need for stability and internal coherence[35–38] and we try to preserve them even when there is no apparent gain[38–40]. This inherent tendency towards stability induces individuals to preferentially enact behaviors aligned with their self-views. In RL terms, this could be seen as a built-in policy space (set of ingrained traits, such as "independence," guiding behavior), where some policies (e.g., prioritizing autonomous actions over collaborative ones) are readily accessible, preferentially activated, and their baselines remain relatively insensitive to environmental changes (e.g., entering a teamwork-oriented culture), opening the door to recurrent mismatches between self-expressions and their estimated utility. These resulting mismatches, if aggregated through correlated experiences and contexts, may result in a relatively stable subjective perception of misfit, capturing the underlying notion of Self-Utility Distance.

Central to this proposal is the principle that Self-Utility Distance reflects an unresolved change signal, akin to a prediction error in RL. In RL, prediction errors signal the difference between predicted and actual outcomes, prompting individuals to update their behavioral strategies. Similarly, Self-Utility Distance might influence Self-Concept Clarity by serving as internal feedback that signals the need for adaptive changes that are difficult to implement for the individual. In line with this conceptualization, research suggests that a perceived need for personal change can undermine the structural integrity of self-knowledge by pressing individuals to adapt to unmatched social demands[41]. Moreover, this perceived need for adaptation can induce self-concept malleability, triggering subtle, unintentional behavioral shifts[42] and foster ambivalence regarding the expression of self-views due to internal incongruities between their current state and perceived necessary changes[15].

To empirically evaluate our proposal, we conducted three behavioral studies. The first study provided an initial test of the hypothesized relationship between Self-Utility Distance and Self-Concept Clarity. In the second study, we conceptually and empirically compared Self-Utility Distance to the components of the Self-Discrepancy Theory. In the third study, we employed computational models to investigate how individuals may strategically adjust their perceived utility estimations when faced with social feedback. Specifically, participants performed a learning task where they learned about socially shared evaluations regarding the utility of various personal characteristics, allowing us to test for different learning strategies that individuals could use to minimize Self-Utility Distance. Together, these complementary studies provide insights into the putative role of Self-Utility Distance in understanding the subjective experience of self-concept clarity.

## Methods
### Study 1. Overview
In this study, we aimed to test the hypothesis that a greater perceived Self-Utility Distance is associated with decreased Self-Concept Clarity.

In exploring the relationship between Self-Utility Distance and Self-Concept Clarity, we also considered the role of Self-Esteem. Self-Esteem has consistently shown a recurrent, moderate to strong correlation with Self-Concept Clarity, suggesting an overlap between the two constructs[12,19,22]. Therefore, by including Self-Esteem, we aimed to assess whether Self-Utility Distance accounts for unique aspects of Self-Concept Clarity beyond those explained by or shared with Self-Esteem. This approach allowed us to test the incremental validity of Self-Utility Distance in relation to one of Self-Concept Clarity's most robust correlates.

Indeed, there is room to expect Self-Utility Distance to account for some of the variance shared between Self-Esteem and Self-Concept Clarity. For example, as long as Self-Esteem includes the perception of an individual's competence and fit with the environment[43,44] Self-Utility Distance and Self-Esteem may similarly capture variations in Self-Concept Clarity levels. However, we expected Self-Utility Distance to share unique variance with Self-Concept Clarity. For example, Self-Esteem is a broad affective measure, representing an individual's global feelings of self-worth and acceptance[45]. In contrast, Self-Utility Distance signals the presence of unresolved tension between one's current self-attributes and perceived environmental incentives for change based on their utility value. This fit does not necessarily align with the individual's emotional well-being. For example, individuals may recognize that their personal characteristics are highly useful in their work environment, even if this environment is stressful or misaligned with their personal preferences. Here, low Self-Utility Distance might be associated with higher Self-Concept Clarity by confirming the utility of one's traits, whereas low Self-Esteem, reflecting discontent with the environment or misalignment with personal values, might negatively affect it.

We hypothesized that Self-Utility Distance would be negatively correlated with Self-Concept Clarity. In addition, we anticipated that this relationship would remain significant after accounting for Self-Esteem, underscoring the unique contribution of Self-Utility Distance in explaining variations in Self-Concept Clarity.

## Study 1. Participants

Prior to the study, we conducted a power analysis using G*Power[46] to determine the required sample size. We aimed to detect a small-to-moderate effect size ($f^2 = 0.08$, alpha = 0.05, 1-B = 0.8). This analysis specifically addressed the expected $R^2$ increase attributable to the inclusion of Self-Utility Distance in a regression model already accounting for Self-Esteem. The results indicated that a minimum of 87 participants would be required. 162 undergraduate students were recruited through the lab panel of the University of Barcelona and were compensated with course credits. Note that the final sample size exceeded the number initially suggested by the power analysis due to technical issues with the university's lab panel. All participants provided informed consent. Similar to a recent study, participants ($n = 7$) missing more than 20% of the responses were excluded from the sample[47]. The final sample was composed of 155 individuals (97 women, 58 men, $M_{age} = 24.07$, $SD_{age} = 7.42$, range = 18–57, participants were asked to report their gender in a multiple-choice question including Woman, Man, Non-Binary participants, "Other" (specify) and "Prefer not to say"). Data was collected between January and March 2024. All studies were approved by the local research ethics committee (University of Barcelona's Bioethics Commission: IRB00003099). Given the small variability in terms of race and ethnicity in participants enrolling from the university's lab panel, this data was not collected. For all studies reported in this research, all parametric tests met statistical assumptions. None of the studies were preregistered.

## Study 1. Procedure

Participants engaged in a task that involved providing both self and utility evaluations for a list of 50 adjectives (see "Stimuli"). The task was divided into two blocks: self-evaluation and utility estimation. The order of the blocks was randomized across participants. In the self-evaluation block, participants rated how well each adjective described them on a scale from 1 (Not at all) to 100 (Perfectly). In the utility estimation block, they assessed how useful they perceived each trait to be for their current lives, using a scale from 1 (Not useful at all) to 100 (Completely useful). Note that, before providing their estimations, participants were introduced to a definition of utility. We instructed them to consider utility as the capacity of each trait to provide them with positive consequences or help them avoid negative consequences in their current life settings. We also instructed them to consider the utility of each trait 'in general', together with a brief example ["For instance, if you encounter the trait 'Ambitious', you need to evaluate whether expressing this trait has the capacity to lead to positive outcomes or generate negative consequences in your life, in general, and as it is right now."]. Next, participants completed the Self-Concept Clarity scale[1] and the Rosenberg Self-Esteem scale[45]. The Self-Concept Clarity Scale consists of 12 items that assess the clarity and definition of an individual's self-concept, such as 'In general, I have a clear sense of who I am and what I am.' Responses are collected using a 5-point Likert scale. Additionally, the Rosenberg Self-Esteem Scale includes 10 items aimed at measuring global self-worth with prompts like 'On the whole, I am satisfied with myself,' utilizing a 4-point Likert scale. The order of presentation of the scales was also randomized across participants.

Self-Utility Distance was quantified as the mean of the absolute differences between self-ratings and utility ratings for each adjective. This method, akin to Manhattan distance, ensures that the measure is normalized for any missing data, thereby maintaining consistency and comparability of Self-Utility Distance scores across all participants. Self-Utility Distance captured the overall dissimilarity between how participants perceived themselves (self-evaluation) and how they assessed the functional utility of their traits within their current life contexts (utility evaluation).

## Study 1. Stimuli

Stimuli consisted of 25 positive and 25 negative traits selected from prior studies[38,47–49], which come from widely studied lists of personality descriptors[50], see Supplementary Table S1. Adjectives were chosen to represent a broad spectrum of personal attributes that individuals might perceive as having varying degrees of utility, such as those included in the HEXACO model of personality[51], together with trait adjectives representing additional dimensions (e.g., 'Authoritarian', 'Practical').

## Study 2. Overview

In Study 1, we introduced Self-Utility Distance, based on a framework that merges cognitive and personality research with reinforcement learning principles, as a computational approach to understanding self-concept clarity. Our findings suggested that the Self-Utility Distance approach is a viable way to understand structural and, potentially, affective self-concept dynamics (Self-Esteem). However, Self-Utility Distance's theoretical roots suggest parallels with Self-Discrepancy Theory, a well-established psychological framework[52,53].

The Self-Discrepancy Theory is a theory of self and affect that delineates various self-representations—namely, the actual self, the ideal self, and the ought self—and suggests that discrepancies among these representations can lead to distinct emotional experiences[52,53]. The actual self includes traits that an individual believes that they possess. In contrast, the ideal and ought selves serve as motivational benchmarks for self-assessment, reflecting their aspirations and perceived duties, respectively. The theory suggests that these self-discrepancies have a wide variety of impacts on individuals' emotional outcomes, potentially contributing to psychopathology[53]. Self-discrepancy research has also explored connections to positive psychological states. For example, existing evidence suggests that lower self-discrepancies relate to higher self-esteem and increased positive affect[52,54,55]. Moreover, although originally defined as a theory to explain affective states, Self-Discrepancy Theory has also shown potential to understand structural components of the self-concept[15].

Both Self-Utility Distance and Self-Discrepancy Theory focus on discrepancies involving individuals' self-concept. In both frameworks, discrepancies signal misalignment. Moreover, both Self-Utility Distance and Self-Discrepancy Theory suggest that these discrepancies are associated with problems in psychological functioning. Self-Discrepancy Theory links such disruptions to emotional states like self-esteem, anxiety or depression, while Self-Utility Distance primarily ties them to structural components of the self-concept. The defining strength of Self-Utility Distance lies in its foundation on utility—a concept inherently computational that involves a subjective estimation of the capacity of self-attributes to maximize rewards or avoid harm in individuals' current life settings. That is, it quantifies their capacity to promote adaptation according to the perceived reward structure of the environment. This computational definition allows to conceptualize Self-Utility Distances much like unresolved prediction errors in reinforcement learning. This grounding gives Self-Utility Distance path to formalize its underlying processes that might be more elusive in abstract frameworks such as the Self-Discrepancy Theory. In turn, its mechanistic definition makes Self-Utility Distance not just a snapshot of misalignment but a traceable outcome of the interaction between self-concept stability and environmental demands. Note that its reliance on computational principles does not imply that it is devoid of subjective components. Both self- and utility ratings reflect personal perceptions, but these perceptions are formalizable within a structured framework.

To further develop the Self-Utility Distance framework, it is crucial to evaluate its effectiveness in predicting measures reflecting self-concept dynamics compared to established theories such as the Self-Discrepancy Theory. This comparison will help determine if Self-Utility Distance can offer additional insights beyond the traditional measures of ideal-self and ought-self discrepancies. Moreover, testing the incremental predictive power of Self-Utility Distance is essential to confirm its potential to improve predictive models for key psychological outcomes.

In this study, we extended the procedure employed in Study 1 to incorporate ideal-self and ought-self discrepancies. Our primary hypothesis was that Self-Utility Distance would provide incremental predictive accuracy above and beyond the components of the Self-Discrepancy Theory in the prediction of Self-Concept Clarity. Moreover, we explored whether Self-Utility Distance could also show incremental validity over the components of the Self-Discrepancy Theory in the prediction of Self-Esteem.

## Study 2. Participants

Prior to the study, we conducted a power analysis to determine the required sample size. We aimed to detect a conservative effect size ($f^2 = 0.025$, alpha = 0.05, 1-B = 0.8). This analysis specifically addressed the expected $R^2$ increase attributable to the inclusion of Self-Utility Distance in a regression model already accounting for Ideal-Self Discrepancy and Ought-Self Discrepancy. The results indicated that a minimum of 309 participants would be required. To account for potential data exclusions due to incomplete participation, we enrolled 344 participants. This precaution ensured that even with a data loss of up to 10%, the effective sample size would not fall below the required threshold of 309 participants. Participants were recruited through the online platform http://www.prolific.com and compensated with 9 pounds per hour for participation (~3 pounds). For this study, we recruited Spanish-speaking participants with an age range of 18–40 years without imposing any geographic restrictions. Although we did not actively collect race/ethnicity data, demographic information provided through Prolific indicated that approximately 80% of the sample self-identified as white. All participants provided informed consent. As in Study 1, participants ($n = 21$) missing more than 20% of the responses were excluded from the sample. The final sample was composed of 323 individuals (160 women, 149 men, 14 not reported, $M_{age} = 29.48$, $SD_{age} = 3.09$, range = 20–41).

## Study 2. Procedure

In Study 2, participants completed a refined version of the adjective evaluation task introduced in Study 1, aimed at operationalizing the components of Self-Discrepancy Theory alongside self and utility assessments. In this version, two additional blocks were added. The Ideal Self block asked participants to rate each adjective by considering how closely it aligned with their personal ideals or aspirations. Specifically, participants were asked to rate how much each adjective represented the person they would like to be, on a scale from 1 (Not at all) to 100 (Perfectly). In the Ought Self Block, participants were asked to rate how much each adjective represented the person they feel they should be, on a scale from 1 (Not at all) to 100 (Perfectly). The order of all blocks was randomized across participants. Next, participants completed the Self-Concept Clarity Scale[1] and the Self-Esteem Scale[45]. In line with Self-Utility Distance, Ideal-Self Discrepancy and Ought-Self Discrepancy were operationalized as Manhattan distances.

## Study 3. Overview

The findings from Studies 1 and 2 highlighted an inverse relationship between Self-Utility Distance and the clarity of individuals' self-concepts. This observation underscores the importance of further investigating how individuals might strategically manage the unresolved change signals configuring Self-Utility Distance. From an adaptive learning perspective, individuals would need to adjust their perceptions of trait utility based on environmental feedback to accurately model their environment[56,57]. That is, they need to map socially shared perceptions of which behaviors are appropriate and effective for achieving available goals in the landscape of their social contexts[58]. However, unrestricted learning of social norms could maximize self-utility distance, thereby increasing the perceived misfit and bolstering a need for personal change. To address this challenge, individuals may employ strategies to balance the need to improve their accurate mapping of the environment with the need to manage increases in Self-Utility Distance.

To investigate the mechanisms that individuals may employ to learn about trait utilities, we formalized a series of computational models reflecting distinct learning strategies. These models span from strategies that

straightforwardly integrate socially shared knowledge about the functional utility of different traits to more complex strategies that help mitigate increases in Self-Utility Distance. For example, to prevent the maximization of Self-Utility Distance, individuals might display a biased sensitivity against social cues that signal the need for personal change. That is, they might display asymmetric learning[49,59], discounting that feedback that would maximize Self-Utility Distance. Alternatively, individuals might use their self-concepts as reference points to promote the alignment of utility-related information with their current self-views. Delineating the underlying social learning mechanisms involved in Self-Utility Distance could enhance our understanding of how individuals manage the change signals that might disrupt the clarity of their self-concepts.

In this study, participants underwent a social learning task where they learned about socially shared perceptions of trait utilities. The task was divided into two blocks. In the first block, participants evaluated their own characteristics using the same set of trait adjectives employed in Studies 1 and 2. In the second block, participants evaluated the utility of the same trait adjectives while receiving trial-by-trial feedback. Through this feedback, participants had the opportunity to learn by adjusting their subsequent trait utility estimations in light of the feedback received. The data resulting from the learning task was used to fit and compare our set of computational models, offering key insights into the mechanisms of trait utility learning that contribute to managing Self-Utility Distance.

## Study 3. Participants

For this study, 92 undergraduate students were recruited through the lab panel of the University of Barcelona and were compensated with course credits. This sample size was based on the largest sample size employed across experiments from prior studies with similar analytical strategies[47,60], plus the addition of 20% of participants to accommodate potential data exclusions. However, due to ongoing issues with the university's lab panel, the actual recruitment slightly exceeded our target sample size ($n = 71$). Seven participants were excluded due to missing more than 20% of the responses during the experimental task. The final sample size was composed of 85 individuals (59 women, 26 men, $M_{age} = 20.97$ years, $SD_{age} = 3.96$ years, range = 18–41, Participants were asked to report their gender in a multiple-choice question including Woman, Man, Non-binary, "Other" (specify) and "Prefer not to say"). All participants provided informed consent. Data was collected between March and June 2024.

## Study 3. Procedure

The experimental task was designed following methodologies established in previous studies[61]. Participants engaged in a social learning task consisting of two separate blocks. In the first block, they provided self-assessments on various trait adjectives (see "Stimuli," Study 1). In the second block, they provided their subjective estimation of the utility of those traits, followed by social feedback consisting of average ratings of trait utilities from a reference group (232 individuals with similar age and educational backgrounds; see Supplementary Materials). In the first block, at the beginning of each trial, participants encountered the prompt "How do you see yourself?" accompanied by an adjective (e.g., 'Sociable'). Below, a slider scale from 1 to 100 was presented. Participants were instructed to rate how much they identified with the trait, with 1 indicating "not at all" and 100 meaning "extremely." In the second block, participants were prompted to evaluate the utility of the same set of traits, responding to the prompt, "How useful do you think this trait is?" They had 15 s to provide their estimation. Right after this estimate, participants received feedback showing the average utility estimations for that trait from the reference group. The feedback appeared on the screen in the format: "Others think the utility of this trait is:" followed by a score ranging from 1 to 100. This score was displayed for 3 s (Fig. 1). This sequence was repeated for all 50 traits involved in the task. Importantly, participants were not explicitly instructed to learn from the feedback. After the task, they completed the Self-Concept Clarity and Self-Esteem scales.

**Fig. 1 | Overview of the experimental task.** During the first block, participants provided self-ratings for a set of 50 traits (e.g., 'Responsible) on a scale from 1 ('not at all') to 100 ('extremely'). In the second block, participants provided their estimations of trait utilities on the same set of traits and received trial-by-trial feedback showing the average utility rating for that trait by a reference group [others] (i.e., 232 psychology undergraduates). The difference between the participant's utility rating and the feedback score represents the Prediction Error (PE). Judgments were separated by inter-trial intervals of 500 ms. This process was iterated for a set of 50 different traits.

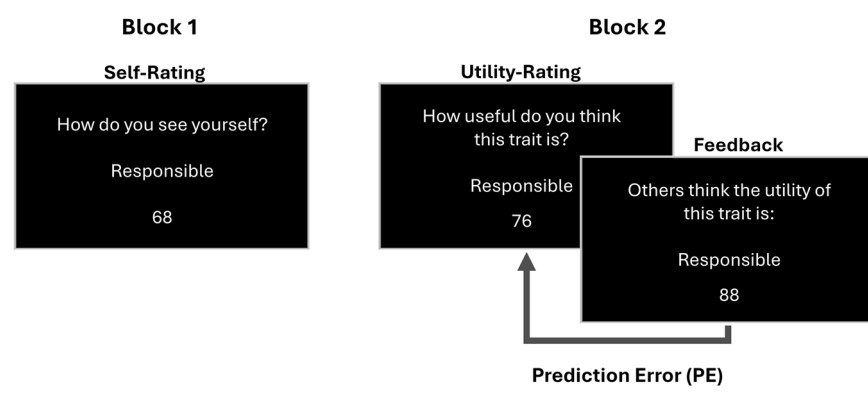

**Computational models.** We formalized five computational models to investigate which model best described participants' learning strategies. Our models were inspired by recent research in learning about others' personalities[47]. This research indicates that when learning about others, participants use fine-grained inter-trait relationships to spread prediction errors and promote learning. This learning mechanism (henceforth, fine granularity) entails the adjustment of expectations for upcoming traits based on the difference between the participant's estimation of a given trait (e.g., 'Responsible') and the feedback received [that is, the prediction error (PE)] via a similarity matrix. For instance, if a participant experiences a prediction error (PE) of '30' for the trait 'Responsible,' this PE will influence the updates of related traits in subsequent evaluations. Suppose 'Responsible' correlates with 'Punctual' 0.5. The update to 'Punctual' would then involve half of the prediction error received for the trait 'Responsible' calculated as $PE_{Responsible} * 0.5 [r_{(Responsible, Punctual)}]$. This adjustment is further shaped by the learning rate, a (free) parameter that quantifies participants' responsiveness to PEs. Similarity matrices, along with feedback ratings, were computed from the ratings provided by a separate group of 232 individuals (see Supplementary Note 1). Four of our five computational models were formalized as hybrid Rescorla Wagner (RW) models, including, but not limited to, a fine granularity learning mechanism. The remaining model consisted of a regression that assumes participants' trait utility estimations derived directly from a linear transformation of self-ratings, representing 'no learning'.

Model 1: No learning. Model 1 assumes that participants perform a linear transformation of their self-ratings (S) to predict (P) trait utility ratings. This model performs as a standard linear regression. $\beta0$ represents the intercept and $\beta1$ the slope.

$$P = \beta0 + \beta1 \cdot S$$

Model 2: Fine granularity. Model 2 employs fine-grained granularity and updates all upcoming utility estimations in each trial according to how similar upcoming traits are to the current item. That is, on a trial-by-trial basis, Model 2 updates utility estimations based on the current PE and the learning rate (LR), and weights the spread of the prediction error to upcoming trials via a similarity matrix (SIM).

$$P(t + 1) = P(1) + \sum_{i=2}^{t-1} \alpha \cdot PE(i) \cdot SIM(i, t + 1)$$

Model 3: Fine granularity (2 learning rates). Model 3 expands Model 2 by incorporating asymmetric learning dynamics by means of two distinct learning rates. One learning rate $\{+\}$ is applied when the feedback (F) received for the current trial reduces the distance between self-ratings and participants' trait utility estimation. That is when $|F - S| < |P - S|$. The other

learning rate $\{-\}$ is applied in the opposite scenario, that is when $|F - S| > |P - S|$. This model accounts for the possibility of differential learning trajectories for feedback that increases or reduces the distance between the current self-concept and the estimations of trait utility.

$$P(t + 1) = P(1) + \sum_{i=2}^{t-1} \alpha\{+, -\} \cdot PE(i) \cdot SIM(i, t + 1)$$

Model 4: Self-adjusted fine granularity. Model 4 expands Model 2 by incorporating self-ratings as a reference point. It operates by combining the self-ratings with the predictions derived from fine granularity learning, employing the free parameter gamma [$\gamma$ (bounded between 0 and 1)] as a balancing factor to weigh the contribution of self-ratings against the learning-based predictions for each trial. This parameter determines how much participants rely on just the learning mechanism from model 2 or their current self-views. For example, if gamma has a value of 0.5, the contribution of the self-concept and learning based on PEs to the final utility estimation is symmetrical.

$$P^m(t + 1) = P(1) + \sum_{i=2}^{t-1} \alpha \cdot PE(i) \cdot SIM(i, t + 1)$$

$$P(t) = S(t) \cdot \gamma + (1 - \gamma) \cdot P^m(t)$$

Model 5: Self-adjusted fine granularity (2 learning rates). This model combines Model 4 with the dual learning rates from Model 3.

$$P^m(t + 1) = P(1) + \sum_{i=2}^{t-1} \alpha\{+, -\} \cdot PE(i) \cdot SIM(i, t + 1)$$

$$P(t) = S(t) \cdot \gamma + (1 - \gamma) \cdot P^m(t)$$

We fitted and compared our computational models within the Hierarchical Bayesian Inference (HBI) framework. The popularity of HBI has increased due to its enhanced robustness and superior precision in parameter estimation and model selection compared to traditional fixed-effect methods[62]. HBI offers several advantages for simultaneous parameter estimation and model comparison. It accounts for the hierarchical structure of the data and treats model identity as a random effect, making model comparisons less susceptible to outliers[62]. HBI employs a hierarchical approach that estimates the population distribution of the model parameters along with the parameters of each individual given the population distribution, regularizing individual parameter estimates. HBI method for model comparison involves estimating the probability of each individual from being generated by each model and using it to weight the effect of

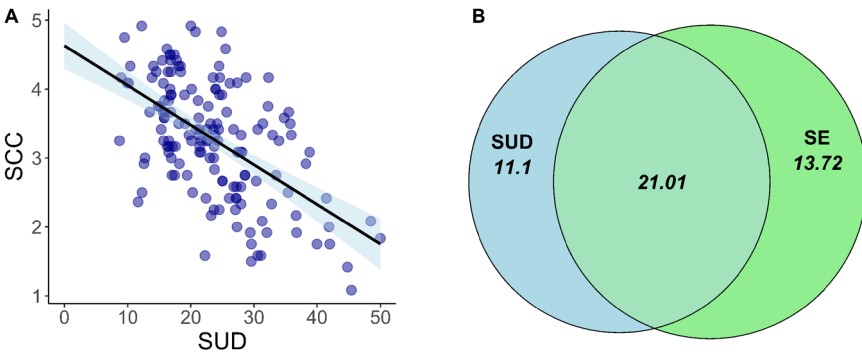

**Fig. 2 | Relationship between Self-Utility Distance and Self-Concept Clarity.** Pearson correlation between Self-Utility Distance (SUD) and Self-Concept Clarity (**A**), the light blue shaded region represents the 95% confidence interval for the regression (n = 155). Commonality analysis, unique and shared explained variance between Self-Utility Distance and Self-Esteem in the prediction of Self-Concept Clarity (**B**).

individual datasets into model fit. It also allows the computation of robust metrics for model comparison and selection, such as the Protected Exceedance Probability (PXP), which is the probability that each model is the most likely across all individuals accounting for the possibility that differences in model evidence are due to chance[62,63]. We fitted our models using the Computational and Behavioral Modeling (CBM) toolbox (https://payampiray.github.io/cbm) implemented in Matlab (Version 2021, a). All models were fitted employing wide Gaussian priors[62]. Initial predictions [P(0)] for models 2 to 5 were set at 80 (on a 1 to 100 scale), establishing a starting point that reflects high expectations toward socially shared perceptions of trait utilities.

## Results
### Study 1. Results
First, we tested the relationship between Self-Utility Distance and Self-Concept Clarity. Consistent with our hypothesis, Self-Utility Distance demonstrated a significant negative correlation with Self-Concept Clarity (r(153) = −0.566, 95% CI [−0.664, −0.449], p < 0.001), indicating that greater Self-Utility Distance is associated with lower clarity in self-concept (Fig. 2). Consistent with prior literature, we also obtained a positive and significant correlation between Self-Concept Clarity and Self-Esteem (r(153) = 0.589, 95% CI [0.475, 0.683], p < 0.001). Moreover, we found a significant negative correlation between Self-Utility Distance and Self-Esteem (r(153) = −0.459, 95% CI [−0.575, −0.325], p < 0.001).

To further explore the unique contribution of Self-Utility Distance to explaining variance in Self-Concept Clarity, we conducted a hierarchical linear regression. In the first model, Self-Concept Clarity was regressed solely on Self-Esteem, which accounted for a significant proportion of variance in Self-Concept Clarity (F(1, 153) = 81.39, p < 0.001, $R^2$ = 0.347). We then added Self-Utility Distance to the model, resulting in an improvement in model fit (F(1,152) = 31.137, p < 0.001, $R^2_{increase}$ = 0.104, $R^2_{adjusted}$ = 0.451), indicating that Self-Utility Distance contributed additional explanatory power beyond Self-Esteem. Standardized regression coefficients indicated a positive effect of Self-Esteem (β = 0.351, SE = 0.057, 95% CI [0.239, 0.463] t(152) = 6.203, p < 0.001) and a negative effect of Self-Utility Distance (β = −0.316, SE = 0.057, 95% CI [−0.428, −0.204] t(152) = −5.580, p < 0.001). Model diagnostics revealed no issues in the multiple regression analysis. Next, we conducted a commonality analysis[64] to partition the unique and non-unique variance explained by model predictors. The results revealed that Self-Utility Distance uniquely accounted for 11.1% of the variance in Self-Concept Clarity, and Self-Esteem uniquely explained 13.7%. Notably, there was a substantial overlap between Self-Utility Distance and Self-Esteem, with both factors together accounting for 21.01% of the common variance in Self-Concept Clarity (Fig. 2). All statistical tests reported are two-sided.

**Control analysis.** Note that, for representing Self-Utility Distance, we decided to use the (average) Manhattan distance (mean absolute difference). This decision was based on its simplicity and the ability to preserve and equally weigh each individual distance between traits and utilities.

This decision was also guided by the absence of specific hypotheses about the relative importance of larger versus smaller distances, which would be differentially weighted in other widely used metrics such as the Euclidean distance. Moreover, we opted against correlation-based measures due to their scale insensitivity, which, although useful in some contexts, does not meet the requirements of our study. Correlation measures focus primarily on the shape and alignment of data without considering the magnitude of discrepancies, which are crucial in our study and central to the operational definition of Self-Utility Distance.

For transparency, we report here the pairwise correlations between Self-Concept Clarity and Self-Utility Distance by using Euclidean distance or Pearson correlations instead of mean absolute differences [note that, in the case of correlation-based measures, the correct interpretation would be SU"S" (similarity)]. Recall that, for the original Self-Utility Distance (mean absolute distance), we found a significant and negative correlation with Self-Concept Clarity (r(153) = −0.566, 95% CI [−0.664, −0.449], p < 0.001). Results suggested similar correlations when Self-Utility Distance was calculated based on Euclidean distance (r(153) = −0.548, 95% CI [−0.649, −0.427], p < 0.001), or Pearson correlations (r(153) = 0.559, 95% CI [0.439, 0.559], p < 0.001), note that for Pearson correlations, the sign of its relationship with Self-Concept Clarity is reversed, representing "similarity."

Our findings supported this hypothesis, revealing a moderate, negative correlation between Self-Utility Distance and Self-Concept Clarity. Crucially, the effect of self-utility distance persisted even after accounting for self-esteem, a well-established predictor of self-concept clarity[12]. By introducing a predictor of Self-Concept Clarity that is grounded in measurable cognitive processes and intersects various domains of psychological research, our study provides a fresh perspective that might enhance understanding of the dynamics of Self-Concept Clarity.

Our findings indicate that Self-Utility Distance and Self-Esteem contribute uniquely and jointly to the explained variance of Self-Concept Clarity. The significant proportion of shared variance between Self-Utility Distance and Self-Esteem may stem from the fact that both reflect aspects of an individual's perceived fit with their environment[43,44]. However, while Self-Esteem is a broad construct, Self-Utility Distance may offer a more specific indicator of the fit between personal characteristics and their perceived functional utility. Notably, Self-Utility Distance and Self-Concept Clarity also shared unique variance, thereby suggesting that Self-Utility Distance might be a tractable and informative indicator to be included in interventions aimed at enhancing Self-Concept Clarity[23].

Note that including Self-Esteem as a predictor of Self-Concept Clarity responded to the aim of establishing Self-Utility Distance as an indicator that provides incremental validity over one of the constructs most recurrently associated with Self-Concept Clarity[12]. However, the directionality of the relationship between Self-Concept Clarity and Self-Esteem has not been robustly established. Including Self-Utility Distance in longitudinal studies could clarify the relationship between Self-Concept Clarity and Self-Esteem[19–22], while also allowing to assess whether Self-Concept Clarity and Self-Esteem reciprocally influence Self-Utility Distance over time.

## Study 2. Results

First, we tested the pairwise relationships between our three main variables (Self-Utility Distance, Ideal-Self Discrepancy and Ought-Self Discrepancy) and the two primary outcomes (Self-Concept Clarity and Self-Esteem). We found that all three variables were significantly and negatively correlated with Self-Concept Clarity (Self-Utility Distance: $r(321) = -0.401$, 95% CI $[-0.489, -0.305]$, $p < 0.001$; Ideal-Self-Discrepancy: $r(321) = -0.351$, 95% CI $[-0.443, -0.252]$, $p < 0.001$; Ought-Self Discrepancy: $r(321) = -0.341$, 95% CI $[-0.434, -0.241]$, $p < 0.001$) and Self-Esteem (Self-Utility Distance: $r(321) = -0.476$, 95% CI $[-0.556, -0.387]$, $p < 0.001$; Ideal-Self-Discrepancy: $r(321) = -0.469$, 95% CI $[-0.550, -0.379]$, $p < 0.001$; Ought-Self Discrepancy: $r(321) = -0.412$, 95% CI $[-0.499, -0.317]$, $p < 0.001$). We also found a positive correlation between Self-Concept Clarity and Self-Esteem ($r(321) = 0.549$, 95% CI $[0.468, 0.621]$, $p < 0.001$) and positive correlations between Self-Utility Distance, Ideal-Self-Discrepancy and Ought-Self Discrepancy ranging from 0.71 to 0.78.

Of primary interest, we tested the unique contribution of Self-Utility Distance to explaining variance in Self-Concept Clarity after accounting for the components from the Self-Discrepancy Theory. As in Study 1, we employed hierarchical regression. In the first model, Self-Concept Clarity was regressed on Ideal-Self-Discrepancy and Ought-Self Discrepancy, which accounted for a significant proportion of variance in Self-Concept Clarity ($F(2, 320) = 26.14$, $p < 0.001$, $R^2_{adjusted} = 0.135$). We then added Self-Utility Distance to the model, resulting in a significant improvement in model fit ($F(1,319) = 10.91$, $p < 0.001$, final model: $F(3,319) = 21.60$, $p < 0.001$, $R^2_{adjusted} = 0.161$). Standardized regression coefficients indicated a negative effect of Self-Utility Distance ($\beta = -0.238$, SE $= 0.072$, 95% CI $[-0.380, -0.096]$, $t(319) = -3.304$, $p = 0.001$) and no significant effects for Ideal-Self-Discrepancy ($\beta = -0.051$, SE $= 0.072$, 95% CI $[-0.214, 0.071]$, $t(319) = -0.703$, $p = 0.482$) and Ought-Self Discrepancy ($\beta = -0.077$, SE $= 0.064$, 95% CI $[-0.174, 0.081]$, $t(319) = -1.194$, $p = 0.233$). Multicollinearity analysis (Variance Inflation Factors, VIF) indicated that the results were not influenced by the correlations between predictors (all VIF $< 3$). Next, we focused on identifying the best-fitting model that incorporates any combination of predictors (Self-Utility Distance, Ideal-Self-Discrepancy and/or Ought-Self Discrepancy), alongside Self-Esteem. We employed the Bayesian Information Criterion (BIC) for model comparison. Results indicated that the best-fitting model was the model that included only Self-Utility Distance and Self-Esteem as predictors of Self-Concept Clarity ($F(2,320) = 77.88$, $p < 0.001$, $R^2_{adjusted} = 0.323$). Standardized regression coefficients indicated a positive effect of Self-Esteem ($\beta = 0.384$, SE $= 0.043$, 95% CI $[0.299, 0.469]$, $t(320) = 8.890$, $p < 0.001$) and a negative effect of Self-Utility Distance ($\beta = -0.149$, SE $= 0.043$, 95% CI $[-0.235, -0.065]$, $p < 0.001$).

To explore whether Self-Utility Distance can be understood as an affective signal similar to the components of the Self-Discrepancy Theory we aimed to reproduce the same analyses but focusing on Self-Esteem as the dependent variable. First, we compared a baseline model including only Ideal-Self Discrepancy and Ought-Self Discrepancy against another including both predictors plus Self-Utility Distance. The initial model, including Ideal-Self Discrepancy and Ought-Self Discrepancy as predictors, was statistically significant ($F(2,320) = 48.53$, $p < 0.001$, $R^2_{adjusted} = 0.227$). We then added Self-Utility Distance to the model, resulting in a significant improvement in model fit ($F(1,319) = 9.25$, $p < 0.001$, final model: $F(3,319) = 36.27$, $p < 0.001$, $R^2_{adjusted} = 0.247$). Standardized regression coefficients indicated a negative effect of Self-Utility Distance ($\beta = -0.167$, SE $= 0.055$, 95% CI $[-0.276, -0.059]$, $t(319) = -3.042$, $p = 0.002$) and Ideal-Self-Discrepancy ($\beta = -0.146$, SE $= 0.055$, 95% CI $[-0.255, -0.037]$, $t(319) = -2.644$, $p = 0.008$). No significant effect was found for Ought-Self Discrepancy ($\beta = -0.052$, SE $= 0.049$, 95% CI $[-0.149, 0.045]$, $t(319) = -1.059$, $p = 0.290$). Finally, we also explored whether Self-Utility Distance might be included in the best-fitting model predicting Self-Esteem. We employed the same model selection approach previously used for Self-Concept Clarity, but with Self-Esteem as the outcome and Self-Concept Clarity as a potential predictor, alongside Self-Utility Distance, Ideal-Self-Discrepancy, and Ought-Self-Discrepancy. The analysis revealed that the best-fitting model incorporated Ideal-Self-Discrepancy and Self-Esteem ($F(2,320) = 101.96$, $p < 0.001$, $R^2_{adjusted} = 0.385$). Standardized regression coefficients indicated a positive effect of Self-Concept Clarity ($\beta = 0.293$, SE $= 0.031$, 95% CI $[0.232, 0.355]$, $t(320) = 9.494$, $p < 0.001$) and a negative effect of Ideal-Self-Discrepancy ($\beta = -0.210$, SE $= 0.031$, 95% CI $[-0.272, -0.149]$, $t(320) = -6.753$, $p < 0.001$). Model diagnostics revealed no issues in the multiple regression analyses. Note that, although Self-Utility Distance was not included in the best-fitting model, it was included in the second best-fitting model (Self-Esteem ~ Self-Utility Distance + Ideal-Self-Discrepancy + Self-Concept Clarity). This suggests that Self-Utility Distance could still have an effect on Self-Esteem. However, this effect might be subtler than that found for the model predicting Self-Concept Clarity.

Our findings suggest that Self-Utility Distance provides incremental validity in the prediction of structural and, potentially, affective components of the self.

One of the central findings of this study is that Self-Utility Distance outperformed Ideal-Self Discrepancy and Ought-Self Discrepancy—the core constructs of Self-Discrepancy Theory in predicting Self-Concept Clarity. While both Ideal-Self-Discrepancy and Ought-Self-Discrepancy were negatively correlated with Self-Concept Clarity, these effects were not significant in a regression model where Self-Utility Distance was included. This suggests that Self-Utility Distance captures unique aspects of self-concept dynamics that are not explained by the Self-Discrepancy Theory. Notably, multicollinearity analysis ruled out the possibility that the shared variance between Self-Utility Distance, Ideal-Self-Discrepancy, and Ought-Self-Discrepancy accounted for these findings, underscoring the distinct predictive power of Self-Utility Distance. Our model comparison analysis further corroborated the central role of Self-Utility Distance in predicting Self-Concept Clarity. When Self-Concept Clarity was the outcome variable, the best-fitting model only included Self-Utility Distance and Self-Esteem as predictors. This finding highlights two important points. First, Self-Utility Distance might provide a more comprehensive understanding of Self-Concept Clarity than the components of Self-Discrepancy Theory, suggesting that self-representational misalignments grounded in functional utility are more relevant to Self-Concept Clarity than those tied to aspirational or normative benchmarks. Second, the inclusion of Self-Esteem in the best-fitting model indicates that affective constructs still play a significant role in self-concept clarity, consistent with prior research on the relationship between Self-Concept Clarity and Self-Esteem[12].

One possible explanation for Self-Utility Distance's superior predictive power lies in its unique operationalization of misalignment (i.e., distance). While the components of the Self-Discrepancy Theory focus on the degree to which self-perceptions diverge from aspirational or normative benchmarks, Self-Utility Distance emphasizes the functional mismatch between self-perceptions and their perceived utility in individuals' current life circumstances. In reinforcement learning, utility is a quantifiable measure of expected cumulative rewards associated with specific states, actions, or decisions. By framing Self-Utility Distance as the discrepancy between self-perceptions and their functional utility, we provide a construct that aligns with the adaptive mechanisms underlying learning processes. As such, Self-Utility Distance measures individuals' perceived "necessary adaptive changes" tied to current self-evaluations, which, akin to modifying behavioral strategies in RL paradigms, might trigger re-evaluation of the current self-structure to match the perceived functional value of self-attributes. In contrast, Ideal-Self-Discrepancy and Ought-Self-Discrepancy, while theoretically rich, lack a comparable mechanistic basis that ties them to measurable learning and adaptation processes. Indeed, ideal or ought views are not specifically tied to current life circumstances and may even necessitate different circumstances to be fully realized. Therefore, these abstract, decontextualized standards might be less likely to reflect change signals capable of affecting the self-concept structure. In line with this notion, our findings also suggest that Self-Utility Distance remains a predictor of Self-Concept Clarity beyond Self-Esteem, indicating that its effect is partially

independent of how closely the self-concept is aligned with its affective status.

We also found that the discrepancy between individuals' self-concept and their ideal self-views predicted self-esteem above and beyond self-concept clarity, consistent with the extensive literature on self-discrepancy theory[52,53]. Here, Self-Utility Distance also showed to be a promising predictor of Self-Esteem; however, our results were not entirely conclusive. While Self-Utility Distance demonstrated incremental validity in predicting Self-Esteem after controlling for the components of the Self-Discrepancy Theory and was included in one of the best models during model selection, it was ultimately excluded from the best-fitting model, which only included Ideal-Self-Discrepancy and Self-Concept Clarity.

One possible explanation is that Self-Utility Distance has a dual effect. First and foremost, it might generate signals indicating necessary changes to better fit the reward structure of the environment, thereby potentially affecting Self-Concept Clarity. Second, similar to prediction errors, Self-Utility Distances may also be aversive to the individual[65], triggering negative emotional responses that may affect Self-Esteem. In turn, these affective responses might help activate regulatory or defensive processes aimed at either adapting behavior or resolving the internal conflict generated by change signals[66]. Critically, the putative effects of Self-Utility Distance on Self-Concept Clarity and Self-Esteem are likely to be interconnected (mirroring the overlap between Self-Concept Clarity and Self-Esteem), with its primary function as a change signal for the self-concept potentially overlapping with its capacity to generate emotional distress. Consequently, when controlling for Self-Concept Clarity, the independent emotional effect of Self-Utility Distance might be subtler and more challenging to isolate, as its affective correlates may be partially entangled with its structural effects. This overlap may explain why its contribution to Self-Esteem appears subtle when Self-Concept Clarity is statistically controlled. In turn, this potentially subtle effect must survive statistical controls for ideal-self discrepancies, which already account for a substantial portion of the variance of Self-Esteem. Future research should specifically target the partial effect of Self-Utility Distance on Self-Esteem to fully unlock its potential as a predictor of affective measures.

## Study 3. Results

Prior to implementing the analysis based on computational models, we conducted a preliminary analysis to assess whether participants learned during the task. We modeled the absolute PEs as a function of time employing a Generalized Additive Model (GAM), which extends traditional linear regression by incorporating smooth functions (Wood, 2017). The results revealed a statistically significant temporal effect on PEs ($F(8.383) = 7.519$, $p < 0.001$), demonstrating a reduction in PE through the course of the task (see Supplementary Fig. S1).

Next, we conducted a Hierarchical Bayesian Inference analysis to determine which computational model best captured participants' responses. Results indicated that the winning model was Model 4 [Self-Adjusted Granularity Model] (model frequency: 89.53%, Supplementary Fig. S1). Further, we computed the Protected Exceedance Probability (PXP), which quantifies the probability that a model is more frequently expressed than any other competing model in the model space while accounting for the possibility that differences in model evidence are due to chance[62,63]. This analysis unequivocally supported Model 4 as the winning model (PXP = 1). Model 4 uniquely integrates the influence of an individual's self-concept on trait utility estimations, employing a hybrid approach that not only incorporates feedback-driven updates but also moderates these updates adjusting them closer to individuals' self-concepts (see "Computational models"). The model's prominence suggests that participants are not only learning from external feedback to align their trait utility estimations with broader social norms but also aligning their learning process with their established self-views. This dynamic suggests a dual process consisting of avoiding the maximization of change signals (SUD) and mapping the utility of personal characteristics. Note that, in our analysis, the gamma parameter in Model 4, which modulates the influence of self-concept versus feedback on learning,

averaged at 0.253 (SD = 0.142). This value suggests that while external feedback predominantly guides participants' updates to trait utilities, the integration of their self-concept remains a notable component of the learning process. By integrating these components, this model provides a comprehensive framework for understanding how individuals learn about socially shared perceptions of trait utility, taking into account both external inputs and internal self-representations. We additionally performed analyses in which Models 2 and 3 were initialized with participants' self-ratings and found that the results remained consistent (see Supplementary Note 3), reinforcing the notion that the effect of individuals' current self-concept parametrized in Model 4 exerts a persistent, potentially motivational influence on the learning process.

To ensure the robustness of our computational models, we conducted parameter recovery analyses demonstrating that our models reliably estimate the true parameter values that generated the data (e.g., Model 4: learning rate r = 0.948, gamma r = 0.996) (Supplementary Fig. S2). Additionally, model distinguishability was confirmed through confusion matrix analysis (see Supplementary Note 2 for details).

Finally, we aimed to test whether our computational parameters α and γ were correlated with measures of self-concept clarity and self-esteem. We found a significant and positive correlation between γ and Self-Concept Clarity (r(81) = 0.345, 95% CI [0.139, 0.521], p = 0.001) and a marginally significant and positive correlation between γ and Self-Esteem (r(81) = 0.198, 95% CI [−0.018, 0.396], p = 0.07). No correlations were found between the learning rate and Self-Concept Clarity or Self-Esteem.

We found that individuals engage in complex computational strategies to adjust their trait utility estimates combining learning from socially shared perceptions of trait utility with their current self-views. The prevalence of this strategy among participants suggests a fundamental motivation to minimize the change signals involved in Self-Utility Distance, which could contribute to avoiding disruptions in the clarity of their self-concepts.

Our findings bridge together two processes extensively studied in psychology: adaptive learning and self-concept stability. On one side, human adaptation necessitates a comprehensive and accurate understanding of the environment, including its available goals, rewards, and dangers[24,27,57,67]. However, individuals' adjustment to perceived environmental demands is to some extent constrained by a tendency for behavioral patterns to cluster around stable baselines[31,32]. Moreover, this tendency is not merely a byproduct of inflexibility, as individuals strive to maintain stable and coherent self-views[35,37,38]. By employing computational strategies that combine adaptive learning and stability preservation mechanisms, individuals can balance the need to accurately map their environments with the need to prevent change signals that could disrupt the stability of their self-concepts.

Our findings also refine research in computational models of social learning[68]. Past research has demonstrated that RL-based computational models can map how individuals learn about others' choices, emotional states, or personalities[69–73]. Here, we demonstrated that when the learning process is potentially motivated (i.e., by the need to reduce Self-Utility Distance), individuals' current self-representations play an important role in structuring the social learning process. Incorporating the self-concept directly into model equations leads to predictions that are not just based on external feedback or generalized learning patterns but are also rooted in individuals' internal structures[68]. This integration provides a more natural characterization of the agent of learning, allowing the parametrization of internal motivations that might conflict with the need to construct an accurate model of the environment.

Building on our findings, we not only extended previous models of social learning but also identified opportunities to merge them with related research. For example, recent studies have explored how individuals update beliefs about themselves, highlighting that some traits are more updatable than others due to their centrality[18,74], a concept borrowed from network theory[75]. Specifically, these studies found that the centrality of a self-belief might influence its susceptibility to change in response to feedback. In this work, the researchers assessed centrality by using subjective estimates of

causal relationships. However, in this research, centrality measures were not included in the computational models as part of the learning generalization mechanism. In contrast, we included traits' interconnectedness directly into our equations, albeit without centrality measures. By integrating these approaches, future research could parametrize centralities as modulators of traits' connectedness influencing feedback spread within computational models. This integration could significantly deepen our understanding of how or whether central traits affect learning processes in response to social feedback. Such an approach could facilitate more granular investigations into the dynamics of the self-concept.

## Discussion

By integrating insights from self-concept dynamics, personality research, and reinforcement learning, we introduced Self-Utility Distance as a predictor that might help illuminate the mechanisms underlying Self-Concept Clarity. In our first study, we found that the unresolved distance between individuals' current self-attributes and their estimated functional utilities is associated with diminished clarity in self-concept. In our second study, we found the stronger and independent predictive power of Self-Utility Distance over Self-Concept Clarity in comparison to the components of the Self-Discrepancy Theory. Finally, in our third study, we provide computational evidence of the underpinnings of the trait-utility learning underlying Self-Utility Distance. Our findings suggest that individuals employ strategies to learn and align socially shared perceptions of trait utility with their current self-concepts, thereby preventing the maximization of Self-Utility Distance in response to environmental feedback. By elucidating the mechanistic principles and predictive capacity of Self-Utility Distance, we provide a fresh perspective that could help clarify the dynamics of Self-Concept Clarity and understand its role as a major predictor of psychological functioning and well-being.

The association found between Self-Utility Distance and Self-Concept Clarity aligns with prior research suggesting that a perceived need for personal change might disrupt the integrity of the self-concept[15,41,42]. Moreover, this finding may offer insights into why variables such as certainty and temporal stability, although central to its definition, do not accurately predict general measures of Self-Concept Clarity[12]. For example, although an individual may be highly certain of their organized, structured, and methodical nature, the estimated functional utility of these traits may be diminished in a new and rapidly evolving work environment (i.e., high Self-Utility Distance). In such instances, holding a strong certainty regarding any personal attribute might not directly translate into subjective Self-Concept Clarity, as those attributes would be perceived as misaligned with the individual's current life circumstances. Additionally, while temporal stability suggests a consistent self-view over time, temporal variations might be the result of both inconsistent self-evaluation and necessary adaptive changes driven by evolving life circumstances[31,76]. We anticipate that those changes that respond to reducing the distance between current self-attributes and their new functional utility in a novel life setting might protect from disruptions in Self-Concept Clarity. In line with this notion, recent research indicates that not all self-concept changes accompanying life transitions disrupt Self-Concept Clarity, as long as they are rewarding for the individual[77].

Importantly, we found that Self-Utility Distance explained unique variance in Self-Concept Clarity and variance common with Self-Esteem. This finding suggests that the link between Self-Esteem and Self-Concept Clarity may partly be due to Self-Esteem's role in enhancing or reflecting perceived environmental fit[43,44]. However, the specific nature of this relationship—whether Self-Concept Clarity shapes, responds to, or synchronizes with Self-Esteem—remains unclear[12,19,20,22]. Incorporating Self-Utility Distance into longitudinal studies could provide insights into these dynamics and test its potentially causal role. For example, such an approach could investigate whether Self-Utility Distance also influences Self-Concept Clarity indirectly through its impact on Self-Esteem. Moreover, the narrower and mechanistic nature of Self-Utility Distance might offer a clearer path for experimental manipulation compared to the broader constructs of

Self-Concept Clarity, Self-Esteem or the components of the Self-Discrepancy Theory.

Our findings can also shed light on the relationships between Self-Concept Clarity and different domains of psychopathology. One remarkable example is the case of the relationship between Self-Concept Clarity and depressive symptoms[78-81]. Individuals with depression often hold highly robust maladaptive self-views, reinforced by cognitive biases[82-85]. This might intuitively suggest a curvilinear relationship between Self-Concept Clarity and depressive symptoms[86], yet such a relationship has not been empirically supported to date. Current findings suggest that while depressive individuals may feel certain about their self-views, this does not necessarily translate into a coherent or stable self-concept. Self-Utility Distance might offer a compelling perspective on this issue. Specifically, in depressive populations, Self-Utility Distance may function as an adaptive signal[87], pressing individuals to consider personal or environmental changes to prevent a further psychological decline. Moreover, the inclusion of Self-Utility Distance in clinical research might also provide important insights into other complex psychopathological phenomena. Specifically, it might help in understanding egosyntonic symptoms—maladaptive perceptions and behaviors that individuals perceive as aligned with their self-concept[88-90]. Such symptoms are notoriously resistant to change, often hindering the efficacy of therapeutic interventions. From our perspective, egosyntonic symptoms could be understood as maladaptive psychological manifestations with high utility for the individual. Incorporating Self-Utility Distance into clinical studies might help delineate the underlying learning mechanisms that sustain these symptoms and impede therapeutic change[91,92].

Beyond the predictive capacity of Self-Utility Distance, our third study elucidated that individuals employ computational strategies that avoid its maximization in response to social feedback. Specifically, our best-performing model indicated that participants tended to align new information about trait utilities with their current self-concept. These findings enhance current computational models of social learning[68] by incorporating the crucial role of the self-concept in the learning process. Incorporating self-concept directly into computational models of social learning allows parametrizing individual differences based on individuals' internal representations, enhancing our understanding of how people engage with and respond to social feedback. Moreover, our computational models might be informative for other research lines. For example, recent research has suggested that despite most individuals perceiving the need to modify some aspects of their self-views[28], intended changes do not always lead to actual changes[29]. To resolve the tension posited by unsuccessful changes, individuals may employ strategies to realign their estimated trait utilities with their current self-concept. Future research might employ our computational models to predict change trajectories and include individual's learning strategies as moderators of the psychological consequences of change failure.

In our operationalization of Self-Utility Distance as a predictor of Self-Concept Clarity, we adopted a generalized approach by assessing trait utilities across individuals' overall life situation. We selected this approach to minimize complexity and provide foundational insights into the relationship between Self-Utility Distance and Self-Concept Clarity. Despite the effectiveness of this approach, it might simplify the ways in which different life contexts—such as work, home, or social interactions—might influence the estimations of traits' utilities. Future research should explore how these context-specific variations might converge within individuals and how might them be weighted into a composite 'general Self-Utility Distance'. Likewise, context-dependent Self-Utility Distances might influence state-like measures of self-concept clarity. Utilizing modern experience sampling methodologies could be particularly effective for this purpose.

To further elucidate the nature and functioning of Self-Utility Distance, it appears beneficial to explore its relationship with well-established error-like signals, such as reward and affective prediction errors (PEs)[93,94]. We defined Self-Utility Distance as an error signal that indicates a necessary adjustment that has not been undertaken by the individual, due to the inherent stability in behavior and self-concept representations. Future

research should investigate whether this error signal or its potential disruptive impact is independent of whether anticipated rewards or emotional states are accurately estimated by the individual.

Finally, to advance our understanding of Self-Utility Distance and its predictive power, future research should also explore which psychological variables underpin its variations among individuals. We propose two potential candidates: Environmental Mastery (EM) and Locus of Control (LOC). EM is defined as the capacity to manage one's environment, make effective use of surrounding opportunities, and choose or create contexts suitable to one's personal characteristics[95]. This capability might translate into reduced Self-Utility Distance by enabling individuals to select or shape their environments in ways that maximize the utility of their personal characteristics. LOC refers to whether individuals attribute life outcomes to their own actions (internal LOC) or external forces (external LOC)[96]. We propose that Self-Utility Distance could be effectively managed by employing a strategic LOC. Specifically, individuals might improve their Self-Utility Distance by externalizing failures (avoiding the maximization of Self-Utility Distance) and attributing successes to themselves (reducing Self-Utility Distance). Future research should investigate this and other individual differences to situate Self-Utility Distance in the landscape of psychological research, potentially refining our understanding of self-concept dynamics.

### Limitations

Building on prior research[18,38,75], we focused on both the content of Self-Utility Distance (studies 1 and 2) and the updating of trait utilities (Study 3) on personal adjectives. However, the self-concept encompasses a wide range of self-representations, including social roles and group memberships. Future studies should explore how the current findings apply to these other aspects of the self-concept. Moreover, we want to highlight methodological consideration (Study 3). Given that feedback ratings were derived from a demographically similar sample and were not manipulated, combined with the low incidence of credibility issues reported in similar studies using manipulated feedback (e.g., ref. 18), we did not assess feedback believability to screen participants. However, this assessment has virtually no cost and might have provided additional information. Future research should include it to ensure best data quality.

### Data availability

Data supporting all studies can be accessed on the Open Science Framework (https://osf.io/6hrzu/)[97].

### Code availability

Code supporting all studies can be accessed on the Open Science Framework (https://osf.io/6hrzu/)[97].

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

## Acknowledgements
This work was supported by the Spanish Ministerio de Ciencia, Innovación y Universidades, which is part of Agencia Estatal de Investigación (AEI), through the project PID2019-111199GB-I00 and PID2022-140426NB-I00 to L.F. (Co-funded by European Regional Development Fund (ERDF), a way to build Europe) and by the German Research Foundation (DFG; specifically by an Emmy Noether Research Group [392443797]) and by the Federal Ministry of Education and Research (BMBF; specifically by a Collaborative Research in Computational Neuroscience (CRCNS) grant) to C.K. We thank CERCA Programme/Generalitat de Catalunya for institutional support. The funders had no role in study design, data collection and analysis, decision to publish or preparation of the manuscript. Funded by AGAUR 2021 SGR 00352.

## Author contributions
J.G.A.: Conceptualization, Investigation, Methodology, Data curation, Formal analysis, Visualization, Writing—Original draft, Writing—Review & Editing. C.W.K.: Conceptualization, Methodology, Writing—Review & Editing, Supervision. L.L.F.: Conceptualization, Writing—Review & Editing, Supervision, Project administration, Funding acquisition.

## Competing interests
The authors declare no competing interests.
