## [Transparent Peer Review file · Communications Psychology]

Self-Utility Distance as a Computational Approach to Understanding Self-Concept Clarity

Corresponding Author: Mr Josué García-Arch

Version 0:

Decision Letter:

Dear Mr Garcí,

Thank you for your patience during the peer-review process. Your manuscript titled "Self-Utility Distance: A Computational Approach to Understanding Self-Concept Clarity" has now been seen by 3 reviewers, whose comments are appended below. You will see that they find your work of some potential interest. However, they have raised quite substantial concerns that must be addressed. In light of these comments, we cannot accept the manuscript for publication, but would be interested in considering a revised version that fully addresses these serious concerns.

We hope you will find the Reviewers' comments useful as you decide how to proceed. Should additional work allow you to address these criticisms, we would be happy to look at a substantially revised manuscript. If you choose to take up this option, please highlight all changes in the manuscript text file, and provide a detailed point-by-point reply to the reviewers.

Editorially, we consider it important to ascertain the conceptual advance of self-utility distance compared to other self-related constructs such as self-discrepancy, on both a theoretical and empirical basis. To this end, we ask that the revision includes an additional experiment that replicates and extends Study 1 to the differentiation of self-utility distance from self-discrepancy. In addition, please either provide additional evidence on the directional relationship between self-utility distance and self-concept clarity or revise the directionality argument between the two concepts in the introduction and discussion. Finally, to protect reviewers' anonymity as part of the peer-review process, please amend the settings on your deposition to grant anonymous access (access without login requirements).

I am attaching a checklist that details critical reporting requirements for the revised manuscript. Please attend to each item and ensure your manuscript is fully compliant. We are requesting that your manuscript aligns with these requirements as this facilitates the evaluation of your manuscript, reducing delays in re-review and potential future acceptance. If your revised manuscript is not aligned with these requests on major issues, such as those concerning statistics, it may be returned to you for further revisions without re-review. Additional information can be found in our style and formatting guide Communications Psychology formatting guide.

If the revision process takes significantly longer than five months, we will be happy to reconsider your paper at a later date, provided it still presents a significant contribution to the literature at that stage.

Please use the following link to submit your
- revised manuscript,

- point-by-point response to the referees' comments,
- cover letter (as a separate document),
- the Editorial Policy Checklist (see below),
- the Reporting Summary (see below), and
- the completed Editorial Request Table (attached):

Link Redacted

Thank you for the opportunity to review your work.

Best regards,

Troby Lui

Troby Lui, PhD
Associate Editor
Communications Psychology

REVIEWER EXPERTISE:

Reviewer #1: reinforcement learning; social learning/cognition

Reviewer #2: reinforcement learning; social learning/cognition

Reviewer #3: self-concept clarity

REVIEWER REPORTS:

Reviewer #1 (Remarks to the Author):

This paper proposes a new approach to understanding self-concept clarity--a social psychological construct important to well-being. Specifically, the authors propose that self-concept clarity depends on "self-utility distance," defined as the distance between one's perceived traits and the utility one perceives those traits to have. They further suggest that people update their beliefs about trait utility in a manner that protects this distance.

There are a number of positive elements in this paper. The paper does a nice job merging social psychological constructs with computational underpinnings; the definition of SUD is sensible and tractable; the idea that people learn about the utility of traits in their environments is interesting and thought-provoking; the approach in Study 2 is a clever way to test learning without having subject re-rate the same items; and the analyses seem appropriate. I think this paper has the potential to be of interest to the field.

At the same time, I do have a few questions I think would be important to address regarding the theorizing, methods, and results.

1) Although the idea of Self-Utility Distance is compelling, I think it would be useful to further address the relations between SUD, self-esteem (SE), and self-concept clarity (SCC). SUD reflects whether one sees one's own traits as valuable, which can also seem like an evaluative judgment consistent with self-esteem. I see the argument that high SUD can serve as a signal for change, and that seeing a discrepancy between one's actual and desired self can promote low self-concept clarity (Demarree & Rios, 2014). At the same time, SUD may also impact self-esteem as a positive/negative evaluation of one's qualities, and further indirectly impact on SCC through this route. Given that part of the motivation stated in the introduction is to gain greater specificity in causal directionality, this seems worth clarifying.

2) On this note, in Study 2, results are not currently reported for any relationships with SCC. This seems like a gap, given that Study 1 results are correlational. Did updated SUD values after learning influence SCC, above and beyond initial values? This would provide a valuable piece of evidence that SUD values play a direct role in shaping SCC (and specifically SCC, as opposed to SE). Similarly, did model parameters relate to SCC?

3) I'd want to hear more about why the authors chose mean absolute difference as a distance measure. I could also see a case for using correlation distance, given that it is scale-insensitive, or Euclidean distance, given that large discrepancies would gain more weight. Intuitively, it also seems as though utility distance might especially matter for traits with extreme utility.

4) On this note, it seems worth considering recent work from Elder et al (2022, 2023a, 2023b) that examined updating of self-concept in response to trait feedback. This work found that (i) subjects resisted change more for traits that other traits depended on (e.g., "witty" is seen as dependent on "fun"), and (ii) subjects high in self-concept clarity had larger learning rates for positive feedback. A key difference between the present work and that work is the nature of the feedback: Elder et al

presented feedback indicating a subject has a trait more or less than previously believed, whereas the present work presents feedback indicating a trait is more or less valued than previously believed. However, that work suggests that some traits are updated more than others, suggesting that trait utility may not matter equally across all traits here as well.

5) In Study 2, how did the average trait utilities presented during feedback compare to participant utilities? For causal inference, it would seem ideal if the trait utilities had been randomly assigned. I recognize this would be challenging here, because participants would be unlikely to believe that very negative traits were rated positively on average by others, so utility could only be manipulated within general valences. I also recognize that there is value in presenting actual true ratings with no deception. At the same time, given that this feedback was not randomly assigned, how did the feedback compare to initial subject ratings?

6) In Models 2 and 3, what values were trait utilities initialized to? I ask because the manuscript describes Model 4 ("self-adjusted fine granularity") as reflecting a motivational strategy, in which individuals avoid maximizing SUD in response to feedback. However, if Model 4 is the only model to combine original self-views and feedback, another possibility is that subjects are rationally updating by combining their prior beliefs and new evidence; if they are very certain of their prior beliefs, they would give low weight to new evidence. In this case, Model 4 would be the only one that can accommodate this kind of combination of new and old information, but it would not require a motivational explanation. Alternatively, Models 2 and 3 could use the subject's initial ratings as a baseline before updating begins.

7) I'm not sure it's quite right to say the mechanisms (versus outcomes) of SCC remain unexplored (p. 2); e.g., past work, including work cited in the paper, has tested the idea that it emerges through a combination of self-verification, in which one wants to know what is true about oneself, and self-enhancement, in which one seeks positive information about the self, leading to contradictory beliefs.

Reviewer #2 (Remarks to the Author):

The authors describe two behavioral studies examining the mechanisms supporting a stable, coherent self-concept. Specifically, they show that "Self-Utility Distance" (the distance between self-ratings and utility ratings for a given trait) predicts self-concept clarity (even when accounting for self-esteem). Further, they illustrate that when learning about consensus-based estimations of trait utility, a model incorporating their baseline self-views best fit the data. Taken together, this work is elegant and timely, and provides an excellent illustration of how computational models can expand our understanding of core self-referential (and social cognitive) processes. I did have a considerable number of questions, suggestions, and requests for clarity, which I hope the authors will be able to attend to.

1. While I appreciate the authors striving for conciseness, at times, they're almost *too* concise. There were stretches of the Introduction that felt like they needed elaboration and grounding. For example, the authors write, "The predictive power of SCC is pervasive across diverse domains of psychological functioning, psychopathology, and well-being," but do not explain the nature of those predictions or domains. This means that when, later, they motivate the current proceedings by referring to "the importance of SCC for understanding different psychological processes," this argument could use more foundation.

On a related note, the authors lean heavily on acronyms – I think in an attempt to meet length requirements. There are big, important ideas here that get obscured a bit by sentences like, "Crucially, the effect of SUD on SCC persisted even after accounting for SE, a well-established predictor of SCC." I found myself wondering if the authors might need more space to tell a more impactful version of this story.

2. Along the same lines, at numerous points, I found myself wanting more explication. For example, the authors write, "Put simply, SUD reflects the distance between current self-attributes and their 'expected utility values'." That's a start, but an illustration of cases that exemplify a high SUD vs. a low SUD would be useful.

Later, the authors write, "In RL terms, this could be seen as a built-in policy space, where some policies are readily accessible, preferentially activated, and their baselines remain relatively insensitive to environmental changes, opening the door to recurrent mismatches between self-expressions and their estimated utility." Again, it would be great to make this concrete through an example framed in terms of real social behavior.

The authors get closer at points – for example, they write that "individuals may recognize that their personal characteristics are highly useful in their work environment, even if this environment is stressful or misaligned with their personal preferences." Even still, I wanted a *bit* more clarity. I'm assuming here that this example would reflect low SUD (i.e., the individual sees their traits as matching the environmental expectations) *and* potentially lower SE (i.e., their behavior is not matching their values, affecting their well-being), and then the question is... what's the consequence for SCC? Is that accurate?

3. I was surprised not to see recent work from Brent Hughes and Jacob Elder cited in the Introduction – in particular, Elder, Davis, & Hughes (2022, "Learning About the Self: Motives for Coherence and Positivity Constrain Learning From Self-Relevant Social Feedback") and Elder, Cheung, Davis, & Hughes (2023, "Mapping the self: A network approach for understanding psychological and neural representations of self-concept structure"). Similar to the present proceedings, this

work examines how representations of the interconnectedness between traits shape learning processes that help construct the self-concept, with consequences for self-concept coherence/clarity. I was hoping that the authors might consider how this work interfaces with their own.

4. I had a few questions regarding the measures and materials the authors employed.

First, the authors explained one input to the SUD measure by saying that participants “assessed how useful they perceived each trait to be for their current lives, using a scale from 1 (Not useful at all) to 100 (Completely useful).” Presumably these evaluations of usefulness will vary across contexts, *and further* the extent to which it *varies* will be different across people. Some people will see the same traits as useful across contexts (and/or may only experience a limited number of contexts routinely), and others might give very different answers for home, work, school, friends, family, etc. How might this be captured and how might it shape the relationship between SCC and SUD?

Second, the authors’ stimuli comprised “25 positive and 25 negative traits selected from prior studies.” Judging based on the supplementary materials, presumably some of the positive (“detail oriented”, “practical”) or negative (“impulsive”, “sarcastic”) traits might be *seen* as having the opposite valence depending on whether people view them as central to their identities – is that fair to say? If so, is valence meaningful here, or were the authors just aiming for a wide variety of traits? (Also, this is a small note, but in the supplement, the traits appear in a *mostly* alphabetic listing, but then that seems to change at points in the list in the supplement – is that intentional?)

Finally, could the authors give sample items and scoring details for the Self-Concept Clarity scale and the Self-Esteem scale?

5. I had a few questions regarding the power analyses, samples, and transparency.

In Exp. 1, the authors note that a power analysis “...indicated that a minimum of 87 participants would be required.” However, they go on to say that “162 undergraduate students were recruited through the lab panel of the University of Barcelona,” with a final sample of 155. It seems like the recruited sample is almost twice as large as the number suggested by the power analysis. How was the stopping rule determined? How and why did the authors decide to run 162 participants instead of 87?

Incidentally, this undergraduate sample has a mean age of 24.07, with a standard deviation of 7.42. This is a bit surprising for an undergrad sample (which typically has a low SD); was this just caused by a handful of folks far outside the typical 18-24 range?

In Exp. 2, we get considerably less detail on the sample size determination. The authors simply write that “[t]his sample size was based on prior studies with similar analytical strategies.” That said, Frolichs et al. (2022; i.e., Ref #44) has five experiments, which comprise 35, 42, 59, 29, and 28 participants, respectively. I’m not seeing how those sample sizes translate to recruiting 92 participants. (Again, how and why was the stopping rule determined?)

Moreover, in addition to the provided detail on age and gender, a race/ethnicity breakdown should be given for both experiments.

Finally, while the authors indicate that “Data and code supporting studies 1 and 2 can be accessed on the Open Science 442 Framework (<https://osf.io/6hrzu/>),” accessing this link requires requesting permissions – which is not ideal given a blind peer review. (Incidentally, as this work is founded quite directly on previous work from these authors, were these experiments preregistered?)

6. Regarding the primary manipulation of utility in Exp. 2 (i.e., “feedback appeared on the screen in the format: “Others think the utility of this trait is:” followed by a score ranging from 1 to 100”), was the believability of this procedure assessed in any way? Were participants screened for suspicion and/or were any exclusion criteria related to believing this feedback employed? If not, this seems like a (minor) limitation.

7. I felt as though the authors went rather quickly through the results, without offering the reader much interpretation or inference. For example, in Experiment 2, the two main findings seem to be that the authors observed “a reduction in PE through the course of the task,” and further, that Model 4 (which “incorporat[es] self-ratings as a reference point” on learning, and uses fine-grained granularity to weight the updating process) best fits the observed data. In other words (I think), a) over time, people are making estimates of trait utility that are closer and closer to consensus perceptions of utility, but b) they’re still biased by their own self-perceptions of utility – in effect, toeing the line between acquiescing to the crowd and staying true to themselves. Perhaps that *specific* inference is off or unwarranted in the authors’ view, but regardless, more scaffolding would be useful here – even before elaboration in the Discussion.

I also thought that the authors could have provided more details on the fine-granularity approach. I had some understanding of this idea from having read Frolichs et al. (2022) before, but I had to go back and review that paper to grasp the models in the present work.

Finally, two additional analyses questions: 1) First, Model 4 included a “free parameter gamma [γ (bounded between 0 and 1)] as a balancing factor to weigh the contribution of self-ratings against the learning-based predictions for each trial.” On average, was gamma skewed towards self-ratings or the learning-based predictions? Did individual differences in gamma

correlated with self-concept clarity or self-esteem? 2) Second, since the authors used positive and negative trait terms (and potentially, people may display asymmetries in how they learn about positive and negative information that's self-relevant), did the authors consider accounting for trait valence in their models? (For example, modeling separate learning rates for positive and negative traits?)

8. Lastly, I noted a few wording issues that could be addressed:

- The authors write, "Yet, research indicates that measures based on these indicators do not accurately predict global indicators of SCC, nor do they exhibit strong correlations among them." The latter part of this sentence might read more clearly as "...nor are they strongly intercorrelated."
- The authors write, "In real-life scenarios, this involves that individuals need to map socially shared perceptions of which behaviors are appropriate..." – the wording of "involves that" is a little confusing; is a word missing perhaps?
- The authors write, "These strategies would allow refining their environmental models 296 while controlling its potential impact on SCC." What is "its" referring to here?
- Finally (and this is less of a wording issue and more a clarity issue), the authors write that they "...demonstrated that when the learning process is motivated (i.e., reduce SUD), individuals' current self-representations play an important role in structuring the social learning process." How is "motivated" being used here? Is the implication that there is a chronic need to reduce SUD? Or was there something in the procedures that explicitly motivated this?

Reviewer #3 (Remarks to the Author):

Review of the paper Self-Utility Distance: A Computational Approach to Understanding Self- Concept Clarity.

The paper introduces the concept of Self-Utility Distance (SUD), which is the absolute difference between the perceived values of specific personal characteristics and their perceived utility. The authors relate SUD to Self-Concept Clarity and, especially in the first study, to Self Evaluation; the authors adopt a functional perspective, where they consider the function of the concept they introduce to the adaptation of the individual to the social environment. In doing so, they adopt the perspective of reinforcement learning, one of the dominant mechanisms in current AI. Two empirical studies are presented as the test of the theory. The first one uses correlational design and multiple regression to examine the relations between SUD, Self-Clarity, and Self-evaluation. The second one is devoted to mechanisms underlying the formation of judgments concerning the utility of traits and examines the role of feedback from peers and perceptions of own traits in estimating the utility of specific traits. After providing their own judgment, the participants were presented with their peers' mean judgments of the utility of traits. The design of the study and the data analysis strategy were similar to the one used by Frolichs (2021), who studied the effects of peers' feedback on the perception of traits of others. Five computational models were used to discover the learning strategy from social feedback.

The paper's theoretical approach is interesting and worth publishing. The analysis of how the Self-Structure regulates behavior to increase adaptation, considering the environmental utility of specific traits, is especially interesting and novel, especially concerning the use of reinforcement learning as the mechanism underlying the formation of the Self-structure. Also, introducing the concept of the utility of self-traits is novel. The concept of the Self-Utility Distance may define a new, important characteristic of the Self-Structure. However, this concept is also close in meaning and measurement to Self-Discrepancy (Higgins 1987). Self-discrepancy is the difference between the actual Self and the ideal Self. The discrepancy can also be measured between the actual Self and the Ought self. The central question to both the proposed theory and empirical studies is how the utility of a trait is related to its ideal value, or the value expected by social norms. This requires both theoretical discussion and an empirical investigation. The authors should explicitly discuss the relation of the Self-Utility Distance to Self-discrepancy. It also might help to replicate the first study measuring Self-discrepancy to examine if the concept of Self-Utility Distance explains unique variance not explained by Self-discrepancy, then the proposed concept presents a unique contribution to the theory of Self. Otherwise, it may be a renaming of one of the primary concepts of the Self, Self-discrepancy. There is considerable literature in psychology that discusses the psychological consequences of Self-discrepancy (e.g. Barnett, Moore, , & Harp, 2017; Higgins, Bond, Klein, R., & Strauman,1986).. Acknowledging this literature would increase the scholarship of the paper. The relationship of the proposed concept to similar existing concepts should be described in more detail.

The concept of utility needs to be better defined because it is a central concept in the proposed model. The similarity of the Self-Utility Distance to existing concepts could also be better judged if the paper in the method section described the exact wording of the instructions to rate the trait utility in more detail. It is crucial because it is unclear if the participants were asked to estimate the importance of each trait for success or what value of this trait is optimal for success.

In sum, the paper is worth publishing if the proposed SUD concept differs from the Self-Discrepancy concept. Answering the question of whether it is different requires a more explicit definition of the utility of traits and a description of rating institutions. An additional study might also be required to examine whether the two concepts can be empirically distinguished.

We ask that you ensure your manuscript complies with our editorial policies and reporting requirements.

To that end, we require revised manuscripts to be accompanied by two completed items: a reporting summary that collects information on study design and procedure, and an editorial policy checklist that verifies compliance with all required editorial policies

- <https://www.nature.com/documents/nr-reporting-summary.zip>>Nature Research Reporting Summary
- <https://www.nature.com/documents/nr-editorial-policy-checklist.pdf>>Editorial Policy Checklist

All points on the policy checklist must be addressed. Your revised manuscript can only be sent back to the referees if these checklists are completed and uploaded with the revision.

Notes: If you have submitted a Stage 1 Registered Report, Review, Primer, Comment, or Perspective you do not need to submit these forms. If you have already submitted these forms, you may disregard this request.

** Visit Nature Research's author and referees' website at <http://www.nature.com/authors>>www.nature.com/authors for information about policies, services and author benefits**

If you experience problems in linking your ORCID, please contact the <http://platformsupport.nature.com/>>Platform Support Helpdesk.

Version 1:

Decision Letter:

Dear Mr García-Arch,

Your manuscript titled "Self-Utility Distance: A Computational Approach to Understanding Self-Concept Clarity" has now been seen by our reviewers, whose comments appear below. In light of their advice I am delighted to say that we are happy, in principle, to publish a suitably revised version in Communications Psychology.

We therefore invite you to revise your paper one last time to address a list of editorial requests. We ask that you edit your manuscript to comply with our format requirements and to maximise the accessibility and therefore the impact of your work.

EDITORIAL REQUESTS:

SUBMISSION INFORMATION:

In order to accept your paper, we require the files listed here <https://www.nature.com/documents/commsj-file-checklist.pdf> .

OPEN ACCESS:

* DATA AVAILABILITY:

Link Redacted

Best regards,

Troy Lui

Troy Lui, PhD
Associate Editor
Communications Psychology

REVIEWERS' COMMENTS:

Reviewer #1 (Remarks to the Author):

The authors have done an excellent job addressing reviewer comments. In particular, the revision is thorough in clarifying the methodological and theoretical approach, along with adding new data and analyses to further clarify the relationships.

I do have one minor suggestion for the authors. In the response letter, they note that they conducted additional analyses in which initial trait utilities in Models 2-3 were based on participants' self-ratings. In these models, the results held constant. I think this is an important finding: it supports the idea that initial ratings have an ongoing and consistent effect--potentially a motivational one--despite new learning, above and beyond setting a starting point for learning. These analyses seem worth mentioning, at least in the supplemental materials. Other than this suggestion, I think the manuscript can be accepted as is.

Reviewer #2 (Remarks to the Author):

The authors have thoughtfully responded to the comments I provided in my initial review. I appreciate their careful revision of this manuscript. While I have a few minor thoughts remaining, in my view, the manuscript is greatly improved. Thank you very much for the opportunity to consider this work!

1. I think this is ultimately a small issue but upon reading the new Study 2 (and rereading Study 1), I think I struggled at times to identify the precise placement/role of certain variables—especially self-esteem—in a logical progression from one to the next. For example, at different times, self-esteem is treated as a competing predictor, a potential confound to be controlled, and an outcome variable. That said, I recognize that maybe my thinking here is a bit reductive and that it's premature to assume a particular serial (or parallel) structure here. Indeed, as the authors state on pg. 9, "...the directionality of the relationship between SCC and SE has not been robustly established. Including SUD in longitudinal studies could clarify the relationship between SCC and SE 19–22, while also allowing to assess whether SCC and SE reciprocally influence SUD over time."

I think ultimately, it would just help to have a line or two motivating the analysis in the new Study 2 that casts SE as the outcome (i.e., "Moreover, we explored whether SUD could also show incremental validity over SUD in the prediction of Self Esteem.") Also, I think that second SUD should be "the SDT measures" or something like that.

2. The authors link SUD to the concept of prediction error several times – i.e., "This computational definition allows to conceptualize SUDs much like unresolved prediction errors in reinforcement learning," "...similar to prediction errors, SUDs may also be aversive to the individual, triggering negative emotional responses that may affect SE," etc. This certainly seems reasonable. Recent work has begun to distinguish between reward (or outcome) prediction errors and *affective* (or emotion) prediction errors (Vollberg & Sander, 2024; Vollberg, & Cikara, 2024; Heffner, Frömer, Nassar, & FeldmanHall, 2024). Can the authors speculate whether the discrepancies captured by SUD map more onto one type of PE vs. the other?

3. In their response letter, the authors referred to several other analyses that they conducted in response to my comments, which they chose not to include in the manuscript (i.e., regarding trait valence and participant age). I agree with their assessments that the analyses are not critical to this work.

Regarding the racial breakdown of their sample, the authors indicated that race/ethnicity is not typically collected, as "over 90% of graduate students in Barcelona are white, which further reflects the homogeneity in the demographic composition of our sample." This makes sense, but I'd recommend simply stating that in text.

That said, the new Study 2 was conducted on Prolific, which does typically yield more racially diverse samples. Did the authors still choose not to collect this from their participants? Moreover, what geographic / nationality / language filters were used when conducting this data collection?

4. Finally, a few typographical errors that I noted:

- pg. 2: "...comprising various personal traits such as 'Sociable' or 'Anxious'" – I think "personal" should be "personality"
- pg. 9: "The actual self includes traits that an individual believes to possess." – should that sentence end with "...believes that they possess"?
- pg. 9: In "The theory suggests that these self-discrepancies have wide variety of impacts on individuals' emotional outcomes..." it looks like "a" is missing between "have" and "wide"
- On pg. 10, "Ought-Self" is misspelled as "Ough-Self"

REVIEWER REPORTS:

Reviewer #1 (Remarks to the Author):

This paper proposes a new approach to understanding self-concept clarity--a social psychological construct important to well-being. Specifically, the authors propose that self-concept clarity depends on "self-utility distance," defined as the distance between one's perceived traits and the utility one perceives those traits to have. They further suggest that people update their beliefs about trait utility in a manner that protects this distance.

There are a number of positive elements in this paper. The paper does a nice job merging social psychological constructs with computational underpinnings; the definition of SUD is sensible and tractable; the idea that people learn about the utility of traits in their environments is interesting and thought-provoking; the approach in Study 2 is a clever way to test learning without having subject re-rate the same items; and the analyses seem appropriate. I think this paper has the potential to be of interest to the field.

At the same time, I do have a few questions I think would be important to address regarding the theorizing, methods, and results.

We thank the reviewer for recognizing the merits of our study, particularly the integration of social psychological constructs with computational underpinnings and the operationalization of Self-Utility Distance (SUD). We are also grateful for the reviewer's enthusiasm about our innovative approach in Study 2. Additionally, we appreciate the reviewer's clear and constructive comments, which have substantially contributed to refining our manuscript.

1) Although the idea of Self-Utility Distance is compelling, I think it would be useful to further address the relations between SUD, self-esteem (SE), and self-concept clarity (SCC). SUD reflects whether one sees one's own traits as valuable, which can also seem like an evaluative judgment consistent with self-esteem. I see the argument that high SUD can serve as a signal for change, and that seeing a discrepancy between one's actual and desired self can promote low self-concept clarity (Demarree & Rios, 2014). At the same time, SUD may also impact self-esteem as a positive/negative evaluation of one's qualities, and further indirectly impact on SCC through this route. Given that part of the motivation stated in the introduction is to gain greater specificity in causal directionality, this seems worth clarifying.

Thank you for your constructive comments and for highlighting the need to clarify the relationships among Self-Utility Distance (SUD), self-esteem (SE), and Self-Concept Clarity (SCC).

The revised manuscript now includes an additional study that tests the incremental predictive power of SUD on SCC and SE, alongside components from the Self-Discrepancy Theory (SDT), which relate closely to evaluative dimensions of the self. In this study, we found that SUD predicts SCC above and beyond the components of the SDT. Conversely, its role in predicting SE was less clear. For example, it showed incremental validity in predicting SE when controlling for the components of the SDT, but was not included in the best fitting model, which included Ideal-Self Discrepancy and Self-Concept Clarity. Following the reviewer's recommendations and the new study's findings, we have revised the manuscript to better address the reviewer's suggestion and strengthen its conceptual focus. For example [lines 537-546]: *One possible*

explanation is that SUD has a dual effect. First and foremost, it might generate signals indicating necessary changes to better fit the reward structure of the environment, thereby potentially affecting SCC. Second, similar to prediction errors, SUDs may also be aversive to the individual⁵⁹, triggering negative emotional responses that may affect SE. In turn, these affective responses might help activating regulatory or defensive processes aimed at either adapting behavior or resolving the internal conflict generated by change signals⁶⁰. Critically, the putative effects of SUD's on SCC and SE are likely to be interconnected (mirroring the overlap between SCC and SE), with its primary function as a change signal for the self-concept potentially overlapping with its capacity to generate emotional distress.

Moreover, we have remarked again the idea suggested by the reviewer in the general discussion [lines 864-865]: *For example, such an approach could investigate whether SUD also influences SCC indirectly through its impact on SE.*

2) On this note, in Study 2, results are not currently reported for any relationships with SCC. This seems like a gap, given that Study 1 results are correlational. Did updated SUD values after learning influence SCC, above and beyond initial values? This would provide a valuable piece of evidence that SUD values play a direct role in shaping SCC (and specifically SCC, as opposed to SE). Similarly, did model parameters relate to SCC?

We thank the reviewer for raising the observation regarding the relationship between models' results and our psychological measures. Similar to prior studies (e.g., Elder et al., 2022) utility expectations were initialized at a predefined value (specified in the methods section as follows: *Initial predictions [P(0)] for models 2 to 5 were set at 80 (on a 1 to 100 scale)*). Note that the choice of initialization values varies across studies. For example, Elder et al. (2022), who modelled how feedback updated beliefs about the self, initialized these values at the midpoint of the scale. In contrast, (e.g.,) Frolichs et al. (2022) modelled them as a free parameter. In our case, we assumed that participants would have positive expectations about trait utilities. We have now explored the validity of this choice by refitting models with initial P as a free parameter, which yielded values near 80 (M = 80.39, SD = 3.41) without improving model fit. Following the reviewer's suggestion we computed "initial SUD" and "updated SUD" and introduced both variables in a regression model predicting SCC. Our results indicated that "updated SUD" had the expected (marginally significant) effect on SCC after controlling for "initial SUD" ($\beta = -.048$, SE = .025, $p = .055$) but not on SE ($\beta = -.011$, SE = .019, $p = .574$). While these findings align with our hypothesis and rationale, we have opted to exclude them from the manuscript due to methodological considerations. Specifically, a robust test of this hypothesis would require both pre and post task measures of self-concept clarity. Moreover, we believe that the brief experimental timeline raises questions about the feasibility of detecting meaningful pre-post task changes in this construct. Nevertheless, we value this empirical direction and will incorporate appropriate temporal measurements in future investigations. We remain open to including these analyses if the reviewer still thinks they are necessary for the current manuscript.

On the other hand, we agree that reporting the correlations between model parameters and SCC is important. Thank you for bringing this to our attention. We have now included this information in the revised version of our manuscript [lines 766-770]: *Finally, we aimed to test whether our computational parameters α and γ were correlated with measures of self-concept clarity and self-esteem. We found a significant and positive correlation between γ and SCC*

($r(81) = .345, p = .001$) and a marginally significant and positive correlation between γ and SE ($r(81) = .198, p = .07$). No correlations were found between the learning rate and SCC or SE.

3) I'd want to hear more about why the authors chose mean absolute difference as a distance measure. I could also see a case for using correlation distance, given that it is scale-insensitive, or Euclidean distance, given that large discrepancies would gain more weight. Intuitively, it also seems as though utility distance might especially matter for traits with extreme utility.

We thank the reviewer for their insightful query regarding our choice of distance measure for representing Self-Utility Distance (SUD). We appreciate the opportunity to clarify our methodology and the rationale behind our specific choices.

In our study, we opted for the mean absolute difference, primarily for its straightforward application and its capacity to treat each distance equally. This choice was guided by our lack of specific hypotheses concerning the relative importance of larger versus smaller discrepancies, as captured by other metrics such as the Euclidean distance. Given the focus of our research on general utility discrepancies across a range of traits, the simplicity of Manhattan distance's linear accumulation of absolute differences was deemed appropriate and aligned with our analytical goals. Regarding the use of correlation-based measures which are scale-insensitive and focus on the shape and alignment of data profiles rather than their magnitude, these were considered less suitable for our specific needs. Our study required a measure that could account for the magnitude of discrepancies, as the operational definition of SUD hinges crucially on the size of the gaps between perceived self and utility ratings. Correlation measures, which normalize the data and focus on patterns of variance rather than absolute differences, would not capture the essence of what our SUD construct aims to reflect.

To ensure a comprehensive analysis and address potential concerns about our choice of distance measure, we have now included control analyses using both Euclidean distance and Pearson correlations. These analyses were meant to test the robustness of our findings across different distance metrics. Importantly, the results were consistent with those obtained using mean absolute distances ($r(153) = -.566, 95\% \text{ CI}[-.664, -.449], p < .001$). Using Euclidean distance, we found a significant negative correlation with SCC similar to that observed with Manhattan distance ($r(153) = -.548, 95\% \text{ CI}[-.649, -.427], p < .001$). Pearson correlations also revealed a significant relationship, albeit with the sign reversed due to its measure of similarity ($r(153) = .559, 95\% \text{ CI} [.439, .559], p < .001$). These findings are now reported in our manuscript for transparency [lines 290-311]. *Note that, for representing SUD, we decided to use the (average) Manhattan distance (mean absolute difference). This decision was based on its simplicity and the ability to preserve and equally weigh each individual distance between traits and utilities. This decision was also guided by the absence of specific hypotheses about the relative importance of larger versus smaller distances, which would be differentially weighted in other widely used metrics such as the Euclidean distance. Moreover, we opted against correlation-based measures due to their scale insensitivity, which, although useful in some contexts, does not meet the requirements of our study. Correlation measures focus primarily on the shape and alignment of data without considering the magnitude of discrepancies, which are crucial in our study and central to the operational definition of SUD.*

For transparency, we report here the pairwise correlations between SCC and SUD by using Euclidean distance or Pearson correlations instead of mean absolute differences [note that, in the case of correlation-based measures, the correct interpretation would be SU"S" (similarity)]. Recall that, for the original SUD (mean absolute distance) we found a significant and negative correlation with SCC ($r(153) = -.566$, 95% CI[-.664, -.449], $p < .001$). Results suggested similar correlations when SUD was calculated based on Euclidean distance ($r(153) = -.548$, 95% CI[-.649, -.427], $p < .001$), or Pearson correlations ($r(153) = .559$, 95% CI[.439, .559], $p < .001$), note that for Pearson correlations, the sign of its relationship with SCC is reversed, representing "similarity".

4) On this note, it seems worth considering recent work from Elder et al (2022, 2023a, 2023b) that examined updating of self-concept in response to trait feedback. This work found that (i) subjects resisted change more for traits that other traits depended on (e.g., "witty" is seen as dependent on "fun"), and (ii) subjects high in self-concept clarity had larger learning rates for positive feedback. A key difference between the present work and that work is the nature of the feedback: Elder et al presented feedback indicating a subject has a trait more or less than previously believed, whereas the present work presents feedback indicating a trait is more or less valued than previously believed. However, that work suggests that some traits are updated more than others, suggesting that trait utility may not matter equally across all traits here as well.

We thank the reviewer for these insightful comments and for drawing attention to the recent work by Elder et al. These observations have helped us consider the implications of our findings in a broader context.

We acknowledge the possibility, as the reviewer suggested, that some trait utilities may be less susceptible to updates than others, similar to the findings by Elder et al., Although our current dataset does not include measures that allow us to distinctly analyze the updatability of trait utilities (e.g., centrality measures distinguishable between in-degree and out-degree), we have included a discussion on this potential variability in trait utility updates. On this note, capturing the differential updatability of trait utilities would require intensive longitudinal data and the construction of temporal/causal networks, which our study design did not originally accommodate. Elder et al. circumvented the need for this data by utilizing participants' estimates of 'subjective causal relationships', which, while ingenious, introduces other complexities in modeling such relationships. Our intuition is that this is the reason why centrality measures in their study were not included in the computational models as part of any generalization mechanism but were instead used in linear mixed-effects models to predict trait re-evaluations (post-feedback phase). This approach, while providing valuable insights, does not integrate these measures into the computational modeling of trait feedback responses directly. That said, we believe that there is a valuable opportunity to integrate the concept of centrality with the advances provided by our computational models. In our study, we included relationships between traits within our computational framework, which, although it does not explicitly account for centrality constraints, does consider the interconnectedness of traits directly into model equations. We have now dedicated a new paragraph to discussing that integrating centrality measures with our computational models could enrich the understanding of self-concept dynamics [lines 806-821]:

Building on our findings, we not only extended previous models of social learning but also identified opportunities to merge them with related research. For example, recent studies have

explored how individuals update beliefs about themselves, highlighting that some traits are more updatable than others due to their centrality^{18,76} a concept borrowed from network theory⁷⁷. Specifically, these studies found that the centrality of a self-belief might influence its susceptibility to change in response to feedback. In this work, the researchers assessed centrality by using subjective estimates of causal relationships. However, in this research, centrality measures were not included in the computational models as part of the learning generalization mechanism. In contrast, we included traits' interconnectedness directly into our equations, albeit without centrality measures. By integrating these approaches, future research could parametrize centralities as modulators of traits' connectedness influencing feedback spread within computational models. This integration could significantly deepen our understanding of how or whether central traits affect learning processes in response to social feedback. Such an approach could facilitate more granular investigations into the dynamics of the self-concept.

As for the relationship between the positive learning rate and SCC found in the mentioned research, together with the overall rationale of their work, we decided to include this reference together with the exceptions to the scarce mechanistic investigations of the dynamics of self-concept clarity, as follows: *...it is essential to build a thorough understanding of the potential mechanisms that drive its formation and maintenance. However, this goal still remains elusive (but see, 15–18).*

5) In Study 2, how did the average trait utilities presented during feedback compare to participant utilities? For causal inference, it would seem ideal if the trait utilities had been randomly assigned. I recognize this would be challenging here, because participants would be unlikely to believe that very negative traits were rated positively on average by others, so utility could only be manipulated within general valences. I also recognize that there is value in presenting actual true ratings with no deception. At the same time, given that this feedback was not randomly assigned, how did the feedback compare to initial subject ratings?

The reviewer raised a thoughtful comment regarding the nature of the feedback presented in Study 2. We opted to use actual ratings based on several considerations: Firstly, to ensure the feedback's believability, and secondly, to effectively utilize the interconnectedness of traits as a generalization mechanism, following the principles outlined by Frolich et al., 2022. While we acknowledge the potential benefits of employing experimentally manipulated feedback, such a strategy poses significant challenges in maintaining credibility. Manipulating feedback within restricted valence boundaries could also limit the naturalistic learning strategies participants might employ, potentially skewing the ecological validity of the findings. Consistent with this notion, we opted to derive feedback ratings from a large, separate sample that shared demographic characteristics with our experimental participants, enhancing the ecological validity of our approach.

In terms of the comparison between the feedback and initial participant ratings, both showed a certain degree of resemblance with feedback ratings, potentially reflecting the fact that feedback ratings came from participants' own population and the learning strategy delineated by our winning model. The average absolute distance between feedback and participants' trait utilities was 10.96 (SD = 2.53) and the average absolute distance between feedback and self-ratings was 24.37 (SD = 5.47). Both participants' utility ratings and self-ratings showed positive correlations with feedback ratings ($p < .001$). For reference, we also computed a linear mixed-effect model predicting participants utility ratings from (1) feedback ratings and (2) self-

ratings, both predictors were statistically significant ($p < .001$). This suggests that while participants considered the feedback from their reference population relevant, they tended to bias their estimates towards their existing self-concepts, which is consistent with our findings from computational modelling.

6) In Models 2 and 3, what values were trait utilities initialized to? I ask because the manuscript describes Model 4 ("self-adjusted fine granularity") as reflecting a motivational strategy, in which individuals avoid maximizing SUD in response to feedback. However, if Model 4 is the only model to combine original self-views and feedback, another possibility is that subjects are rationally updating by combining their prior beliefs and new evidence; if they are very certain of their prior beliefs, they would give low weight to new evidence. In this case, Model 4 would be the only one that can accommodate this kind of combination of new and old information, but it would not require a motivational explanation. Alternatively, Models 2 and 3 could use the subject's initial ratings as a baseline before updating begins.

We thank the reviewer for this interesting suggestion. As mentioned in our previous response, initial expectations were initialized at the value 80. Please refer to our previous response for details. To explore the interesting possibility the reviewer suggested, we conducted an additional computational analysis. Here, initial values for trait utilities in Models 2 and 3 were based on participants' self-ratings. We performed the same model fitting and comparison procedure as employed in our main analysis (HBI), from which Model 4 still emerged as the winning model (Model frequency 91.5%, Protected Exceedance Probability = 1). This outcome further supports our interpretation that Model 4's integration of self-concept with feedback might reflect a motivational process aimed to aligning external trait utilities with internal self-images.

Moreover, in addressing the nature of Model 4, we think that it is important to highlight that if one wants to consider it as indicative of integrating prior evidence with new information, it is also necessary to acknowledge that the 'prior evidence' in question would be the participants' self-concepts—personal constructs which, rationally, should not directly correlate with estimations of traits' utilities. We believe that this integration is intrinsically self-serving; it is not just an objective updating of utilities based on external feedback but a modulation to maintain consistency with established self-views. In our view, the validity of this assumption was initially supported by findings from our first study, where self-utility distance was significantly correlated with self-concept clarity and self-esteem. For the learning experiment, we now added the correlation between both SCC and SE and gamma, the parameter balancing self-concept with utility updates, and also found positive correlations. In our view, this adds another layer of validation to our argument, emphasizing the self-serving nature of Model 4. Thus, we believe that even if viewed through a lens of rational updating, the inherent motivation to preserve self-concept integrity seems to shape this process.

7) I'm not sure it's quite right to say the mechanisms (versus outcomes) of SCC remain unexplored (p. 2); e.g., past work, including work cited in the paper, has tested the idea that it emerges through a combination of self-verification, in which one wants to know what is true about oneself, and self-enhancement, in which one seeks positive information about the self, leading to contradictory beliefs.

We recognize that literature on self-verification and self-enhancement does include discussions that may suggest mechanisms influencing self-concept dynamics, which intersect

with ideas surrounding SCC. These theoretical frameworks indeed highlight processes through which individuals may affirm or enhance their self-views, contributing to either cohesive or contradictory self-beliefs. These processes indirectly relate to SCC and provide understanding of self-concept dynamics. However, our statement regarding the mechanisms being "unexplored" was aimed at highlighting the relative lack of more direct, specific tests and formal mechanisms proposed for elucidating SCC, particularly when compared to the more frequently discussed 'outcomes' associated with SCC. In our view, while the ongoing debates between self-verification and self-enhancement theories offer valuable insights, they often involve broad constructs and have not yet converged on clear mechanistic definitions or mechanisms that address the specificity of SCC. For clarity, we have adjusted our language from "unexplored" to "underexplored." Note that we also provided references pointing to some exceptions in the original manuscript: *However, this goal still remains elusive (but see, 15–17). (now 15–18).*

Reviewer #2 (Remarks to the Author):

The authors describe two behavioral studies examining the mechanisms supporting a stable, coherent self-concept. Specifically, they show that "Self-Utility Distance" (the distance between self-ratings and utility ratings for a given trait) predicts self-concept clarity (even when accounting for self-esteem). Further, they illustrate that when learning about consensus-based estimations of trait utility, a model incorporating their baseline self-views best fit the data. Taken together, this work is elegant and timely, and provides an excellent illustration of how computational models can expand our understanding of core self-referential (and social cognitive) processes. I did have a considerable number of questions, suggestions, and requests for clarity, which I hope the authors will be able to attend to.

We thank the reviewer for their positive evaluation of our work and for appreciating the rigor and relevance of our approach to understanding self-concept clarity. We are grateful for the recognition of how our use of computational models deepens the understanding of core self-referential and social cognitive processes. Additionally, we appreciate the detailed and constructive feedback provided, which has helped us enhance the clarity and depth of our manuscript.

1. While I appreciate the authors striving for conciseness, at times, they're almost **too** concise. There were stretches of the Introduction that felt like they needed elaboration and grounding. For example, the authors write, "The predictive power of SCC is pervasive across diverse domains of psychological functioning, psychopathology, and well-being," but do not explain the nature of those predictions or domains. This means that when, later, they motivate the current proceedings by referring to "the importance of SCC for understanding different psychological processes," this argument could use more foundation.

We thank the reviewer for recognizing our effort to strike a balance between conciseness and explanation. We appreciate the identification of specific points in the text where this balance could be further refined, such as the need to elaborate more on the predictive power of SCC's domains. We have now expanded the text as follows:

[Lines 52-55]: *For example, higher SCC has been associated with well-being and psychological adjustment³⁻⁵, relationship quality⁶, problem-solving during social conflict⁷, educational achievement⁸, occupational success⁹, reduced job burnout¹⁰, and better mental health^{1,2,11-14}*

On a related note, the authors lean heavily on acronyms – I think in an attempt to meet length requirements. There are big, important ideas here that get obscured a bit by sentences like, “Crucially, the effect of SUD on SCC persisted even after accounting for SE, a well-established predictor of SCC.” I found myself wondering if the authors might need more space to tell a more impactful version of this story.

We appreciate the reviewer’s feedback regarding the use of acronyms in our manuscript. We agree that minimizing acronyms can improve clarity and accessibility. We have revised the manuscript to spell out full terms or use descriptive expressions, particularly in the introduction and discussion sections. Acronyms have been retained in the results section and where concepts are frequently referenced to maintain readability. We are open to making further adjustments if needed.

2. Along the same lines, at numerous points, I found myself wanting more explication. For example, the authors write, “Put simply, SUD reflects the distance between current self-attributes and their ‘expected utility values’.” That’s a start, but an illustration of cases that exemplify a high SUD vs. a low SUD would be useful.

We followed the reviewer’s advice, and we have now included an example of high and low SUD [lines 83-96]:

As an example of high Self-Utility Distance, consider an individual who identifies strongly as being "independent" (e.g., tends to be self-reliant, tends to work autonomously, likes to do plans by herself). If this person is part of a work culture that heavily emphasizes teamwork and collaborative processes (e.g., frequent team meetings, shared projects, group work), they might perceive a low utility in their independence, seeing it as less conducive to gaining rewards (e.g., team-based bonuses, promotions) or avoiding negative outcomes (e.g., job loss). Conversely (low SUD), If the individual works in a setting that values autonomous work (e.g., remote work, flexible project choices), their independent nature would align closely with the environment’s demands. In this context, the individual might perceive high utility in their independence, as it might enhance their ability to achieve rewards (e.g., recognition for individual contributions, opportunities for self-directed projects) and prevent negative outcomes (e.g., conflicts over team roles).

Later, the authors write, “In RL terms, this could be seen as a built-in policy space, where some policies are readily accessible, preferentially activated, and their baselines remain relatively insensitive to environmental changes, opening the door to recurrent mismatches between self-expressions and their estimated utility.” Again, it would be great to make this concrete through an example framed in terms of real social behavior.

We thank the reviewer for this suggestion. As reported in our previous response, we have provided new examples of how self-utility discrepancies might manifest in real social behavior that appear before the mentioned sentence. This particular sentence was intended as a succinct "closing analogy" rather than the introduction of a new concept or premise.

To ensure clarity and maintain the flow, we've added brief, illustrative details directly into the analogy itself [lines 128-132]: “In RL terms, this could be seen as a built-in policy space (*set of ingrained traits, such as “independence”, guiding behavior*), where some policies (e.g.,

prioritizing autonomous actions over collaborative ones) are readily accessible, preferentially activated, and their baselines remain relatively insensitive to environmental changes (e.g., entering a teamwork-oriented culture), opening the door to recurrent mismatches between self-expressions and their estimated utility.”

The authors get closer at points – for example, they write that “individuals may recognize that their personal characteristics are highly useful in their work environment, even if this environment is stressful or misaligned with their personal preferences.” Even still, I wanted a *bit* more clarity. I’m assuming here that this example would reflect low SUD (i.e., the individual sees their traits as matching the environmental expectations) *and* potentially lower SE (i.e., their behavior is not matching their values, affecting their well-being), and then the question is... what’s the consequence for SCC? Is that accurate?

We thank the opportunity to clarify the point raised by the reviewer. In response, we have made a specific clarification in our manuscript to better articulate the independent contributions of Self-Utility Distance (SUD) and Self-Esteem (SE) to Self-Concept Clarity (SCC). We added the following text to the referred sentence [lines 188-190]: *Here, low SUD might be associated with higher SCC by confirming the utility of one's traits, whereas low SE, reflecting discontent with the environment or misalignment with personal values, might negatively affect it.* Please note that the primary intention behind the original example was to illustrate that Self-Utility Distance (SUD) and Self-Esteem (SE) might independently influence Self-Concept Clarity (SCC). This approach was taken to move beyond typically using SE merely as a control variable, which is often based solely on its established relationship with SCC, and to more clearly delineate the distinct contributions of these constructs.

3. I was surprised not to see recent work from Brent Hughes and Jacob Elder cited in the Introduction – in particular, Elder, Davis, & Hughes (2022, “Learning About the Self: Motives for Coherence and Positivity Constrain Learning From Self-Relevant Social Feedback”) and Elder, Cheung, Davis, & Hughes (2023, “Mapping the self: A network approach for understanding psychological and neural representations of self-concept structure”). Similar to the present proceedings, this work examines how representations of the interconnectedness between traits shape learning processes that help construct the self-concept, with consequences for self-concept coherence/clarity. I was hoping that the authors might consider how this work interfaces with their own.

Thank you for highlighting the work of Jacob Elder and Brent Hughes, which we agree should have been cited in the text. We thank the reviewer for the opportunity to including it now. Accordingly, and in line with a similar suggestion made by Reviewer 1, we have dedicated a new paragraph to elaborate on this [lines 806-821]:

Building on our findings, we not only extended previous models of social learning but also identified opportunities to merge them with related research. For example, recent studies have explored how individuals update beliefs about themselves, highlighting that some traits are more updatable than others due to their centrality^{18,76} a concept borrowed from network theory⁷⁷. Specifically, these studies found that the centrality of a self-belief might influence its susceptibility to change in response to feedback. In this work, the researchers assessed centrality by using subjective estimates of causal relationships. However, in this research, centrality measures were not included in the computational models as part of the learning

generalization mechanism. In contrast, we included traits' interconnectedness directly into our equations, albeit without centrality measures. By integrating these approaches, future research could parametrize centralities as modulators of traits' connectedness influencing feedback spread within computational models. This integration could significantly deepen our understanding of how or whether central traits affect learning processes in response to social feedback. Such an approach could facilitate more granular investigations into the dynamics of the self-concept.

4. I had a few questions regarding the measures and materials the authors employed.

First, the authors explained one input to the SUD measure by saying that participants “assessed how useful they perceived each trait to be for their current lives, using a scale from 1 (Not useful at all) to 100 (Completely useful).” Presumably these evaluations of usefulness will vary across contexts, *and further* the extent to which it *varies* will be different across people. Some people will see the same traits as useful across contexts (and/or may only experience a limited number of contexts routinely), and others might give very different answers for home, work, school, friends, family, etc. How might this be captured and how might it shape the relationship between SCC and SUD?

We thank the reviewer for this insightful comment regarding the variability of trait utilities across different contexts. In our current study, the assessment of trait utility was indeed simplified and context-general. Participants were asked to evaluate the utility of traits considering their overall life situation rather than specific contexts like home, work, or social settings. This approach was chosen to avoid complexity and to establish a baseline understanding of the general relationship between SUD and SCC. We consider the potential variability in trait utility across different life domains and its potential implications very interesting. This variability could indeed provide additional understanding of the relationship between SUD and SCC. For example, illuminating their short-term, context-specific dynamics. At the same time, one could argue that despite the variations in utilities across contexts, these may be integrated by individuals into a form of "general SUD". This integrated measure may still meaningfully relate to SCC, as (following the same logics) it would represent a weighted average of context-specific utilities, reflecting a general perception of trait utility across various life domains. We consider that this is an interesting idea and have outlined it in the General Discussion section [lines 908-918]:

In our operationalization of Self-Utility Distance as a predictor of Self-Concept Clarity, we adopted a generalized approach by assessing trait utilities across individuals' overall life situation. We selected this approach to minimize complexity and provide foundational insights into the relationship between SUD and SCC. Despite the effectiveness of this approach, it might simplify the ways in which different life contexts—such as work, home, or social interactions—might influence the estimations of traits' utilities. Future research should explore how these context-specific variations might converge within individuals and how might them be weighted into a composite 'general SUD'. Likewise, context-dependent SUDs might influence state-like measures of self-concept clarity. Utilizing modern experience sampling methodologies could be particularly effective for this purpose.

Second, the authors' stimuli comprised “25 positive and 25 negative traits selected from prior

studies.” Judging based on the supplementary materials, presumably some of the positive (“detail oriented”, “practical”) or negative (“impulsive”, “sarcastic”) traits might be *seen* as having the opposite valence depending on whether people view them as central to their identities – is that fair to say? If so, is valence meaningful here, or were the authors just aiming for a wide variety of traits? (Also, this is a small note, but in the supplement, the traits appear in a *mostly* alphabetic listing, but then that seems to change at points in the list in the supplement – is that intentional?)

Thank you for your observations regarding the selection and organization of the traits used in our study. We indeed aimed to include a broad spectrum of traits to capture a diverse range of self-concept attributes that individuals might perceive as having varying degrees of utility. We added a brief sentence to reflect this in the manuscript: *Adjectives were chosen to represent a broad spectrum of personal attributes that individuals might perceive as having varying degrees of utility.* Additionally, we appreciate the reviewer’s note on the alphabetical ordering of the traits in the supplementary materials. This minor inconsistency was an oversight, which has been corrected to maintain a consistent alphabetical order.

Finally, could the authors give sample items and scoring details for the Self-Concept Clarity scale and the Self-Esteem scale?

Of course. We have now included this information for both the Self-Concept Clarity scale and the Self-Esteem scale in the manuscript [lines 236-241]:

The Self-Concept Clarity Scale consists of 12 items that assess the clarity and definition of an individual's self-concept, such as 'In general, I have a clear sense of who I am and what I am.' Responses are collected using a 5-point Likert scale. Additionally, the Rosenberg Self-Esteem Scale includes 10 items aimed at measuring global self-worth with prompts like 'On the whole, I am satisfied with myself,' utilizing a 4-point Likert scale.

5. I had a few questions regarding the power analyses, samples, and transparency.

In Exp. 1, the authors note that a power analysis “...indicated that a minimum of 87 participants would be required.” However, they go on to say that “162 undergraduate students were recruited through the lab panel of the University of Barcelona,” with a final sample of 155. It seems like the recruited sample is almost twice as large as the number suggested by the power analysis. How was the stopping rule determined? How and why did the authors decide to run 162 participants instead of 87?

We thank the reviewer for highlighting the need to clarify this important issue. We typically include a buffer of approximately 15-30% (depending on feasibility and resources) more participants than suggested by the initial power analysis. This precaution helps account for any potential incomplete data or data exclusions. In this study, we also used the University’s lab panel, which had recently introduced an option for researchers to set a target number of participants. Regrettably, we did not realize initially that reaching this target would not automatically stop the recruitment process. By the time we reviewed the recruitment progress, the number had already exceeded our initial target. We apologize for this oversight. However, once the data was collected, we decided to proceed with the study as planned, given that the additional data were collected under the same conditions and procedures. We have clarified this issue to prevent any confusion for readers, as noted by the reviewer, as follows [lines 206-208]:

Note that the final sample size exceeded the number initially suggested by the power analysis due to technical issues with the university's lab panel.

Incidentally, this undergraduate sample has a mean age of 24.07, with a standard deviation of 7.42. This is a bit surprising for an undergrad sample (which typically has a low SD); was this just caused by a handful of folks far outside the typical 18-24 range?

Thank you for pointing out the age range in our undergraduate sample. The reviewer was right, the standard deviation is larger than usual because of a few older participants (e.g., 12 participants above 30 years old, among which 5 were between 55 and 57 years old). In contrast, the most typical age in our sample was 20. For transparency we have reported the age range for all studies in our manuscript. We have also checked whether including age as a regressor in our model (SCC as a function of SUD and SE) would change the results. We found virtually no differences in model coefficients sign, magnitude or significance. Additionally, age was not significant ($p = .485$). We believe the inclusion of this analysis in the manuscript is not needed. However, we are open to including it if the reviewer deems it necessary.

In Exp. 2, we get considerably less detail on the sample size determination. The authors simply write that “[t]his sample size was based on prior studies with similar analytical strategies.” That said, Frolichs et al. (2022; i.e., Ref #44) has five experiments, which comprise 35, 42, 59, 29, and 28 participants, respectively. I’m not seeing how those sample sizes translate to recruiting 92 participants. (Again, how and why was the stopping rule determined?)

Thank you for your inquiry regarding the sample size in Experiment 2. We initially aimed to recruit approximately 71 participants, based on the largest sample size from Frolichs et al. (2022), plus an additional 20% to account for potential data exclusions. However, due to ongoing issues with the university lab panel, which had not been fully resolved from the previous experiment, we once again exceeded this target, reaching 92 participants once we recognized the oversight. We appreciate your understanding. We have also clarified this issue, as follows [lines 598-604]:

For this study, 92 undergraduate students were recruited through the lab panel of the University of Barcelona and were compensated with course credits. *This sample size was based on the largest sample size employed across experiments from prior studies with similar analytical strategies^{47,65} plus the addition of 20% of participants to accommodate potential data exclusions. However, due to ongoing issues with the university's lab panel, the actual recruitment slightly exceeded our target sample size ($n = 71$).*

Note that, in the additional study we have included in the manuscript, and given the low rate of data loss in Studies 1 and 2 (now 3), we decided to be less conservative and increase the sample size only a 10% to prevent falling below the estimation provided by the power analysis.

Moreover, in addition to the provided detail on age and gender, a race/ethnicity breakdown should be given for both experiments.

Thank you for your suggestion regarding the inclusion of a race/ethnicity breakdown for both experiments. However, it is important to note that, in our country, this information is not typically collected during research involving university students. In general, over 90% of graduate students in Barcelona are white, which further reflects the homogeneity in the

demographic composition of our sample. We understand the importance of considering diverse perspectives in research, and we will explore ways to ensure inclusivity in future studies while maintaining participant confidentiality and ethical research practices.

Finally, while the authors indicate that “Data and code supporting studies 1 and 2 can be accessed on the Open Science 442 Framework (<https://osf.io/6hrzu/>),” accessing this link requires requesting permissions – which is not ideal given a blind peer review. (Incidentally, as this work is founded quite directly on previous work from these authors, were these experiments preregistered?)

We apologize for any inconvenience caused by the restricted access to the data and code on the Open Science Framework. We have now ensured that it is publicly available. Regarding the preregistration of our experiments, these studies were not preregistered. We now state this explicitly in the paper, for transparency.

6. Regarding the primary manipulation of utility in Exp. 2 (i.e., “feedback appeared on the screen in the format: “Others think the utility of this trait is:” followed by a score ranging from 1 to 100”), was the believability of this procedure assessed in any way? Were participants screened for suspicion and/or were any exclusion criteria related to believing this feedback employed? If not, this seems like a (minor) limitation.

Thank you for your observation. We did not formally assess the believability of the feedback during the experiment. Note that, as in (most experiments from) Frolichs et al. (2022) The feedback employed was veridical. As reported in our manuscript, it was based on data collected from a large sample of university students similar to our target demographic. Moreover, our experience with similar feedback experiments suggests that participants generally find such feedback believable, even if it is slightly manipulated. Similarly, as an example, Elder et al (2022) employed manipulated feedback, and only two participants were excluded using believability as a criterion. Given this, we did not find this assessment necessary. However, we acknowledge that explicitly assessing participants' perceptions of the feedback's believability has virtually no cost and would have provided additional information. We will incorporate this assessment in future studies.

We decided to report this in the manuscript, for transparency [lines 938-948]: *Building on prior research^{18,38,47–49,77} we focused both the content of SUD (studies 1 and 2) and the updating of trait utilities (Study 3) on personal adjectives. However, the self-concept encompasses a wide range of self-representations, including (e.g.) social roles and group memberships. Future studies should explore how the current findings apply to these other aspects of the self-concept. Moreover, we want to highlight methodological consideration (Study 3). Given that feedback ratings were derived from a demographically similar sample and were not manipulated, combined with the low incidence of credibility issues reported in similar studies using manipulated feedback (e.g.,¹⁸) we did not assess feedback believability to screen participants. However, this assessment has virtually no cost and might have provided additional information. Future research should include it to ensure best data quality.*

7. I felt as though the authors went rather quickly through the results, without offering the reader much interpretation or inference. For example, in Experiment 2, the two main findings seem to be that the authors observe “a reduction in PE through the course of the task,” and

further, that Model 4 (which “incorporat[es] self-ratings as a reference point” on learning, and uses fine-grained granularity to weight the updating process) best fits the observed data. In other words (I think), a) over time, people are making estimates of trait utility that are closer and closer to consensus perceptions of utility, but b) they’re still biased by their own self-perceptions of utility – in effect, toeing the line between acquiescing to the crowd and staying true to themselves. Perhaps that *specific* inference is off or unwarranted in the authors view, but regardless, more scaffolding would be useful here – even before elaboration in the Discussion.

Thank you for your feedback regarding the presentation of our results in Study 3. We have expanded this section to provide more detailed findings.

Regarding your observation of the two main findings: the first result indicating a "reduction in Prediction Error (PE) through the course of the task" serves as a preliminary analysis to establish that learning occurred, which is a standard yet very broad check in studies involving computational models. The second finding, which you noted as capturing both the convergence towards consensus utility perceptions and the influence of personal self-concept, indeed subsumes the first. In the current version of the manuscript, we have included additional explanation of the results and additional data such as model frequency of our winning model (Model 4) and the mean value of the gamma parameter (as requested below in a separate comment) The new version reads as follows [lines 736-758]:

Next, we conducted a Hierarchical Bayesian Inference analysis to determine which computational model best captured participants’ responses. Results indicated that the winning model was Model 4 [Self-Adjusted Granularity Model] (model frequency: 89.53 %, Figure S1). Further, we computed the Protected Exceedance Probability (PXP), which quantifies the probability that a model is more frequently expressed than any other competing model in the model space while accounting for the possibility that differences in model evidence are due to chance^{67,68}. This analysis unequivocally supported Model 4 as the winning model (PXP = 1). Model 4 uniquely integrates the influence of an individual’s self-concept on trait utility estimations, employing a hybrid approach that not only incorporates feedback-driven updates but also moderates these updates adjusting them closer to individuals’ self-concepts (see Computational Models). The model’s prominence suggests that participants are not only learning from external feedback to align their trait utility estimations with broader social norms but also aligning their learning process with their established self-views. This dynamic suggests a dual process consisting in avoiding the maximization of change signals (SUD) and mapping the utility of personal characteristics. Note that, in our analysis, the gamma parameter in Model 4, which modulates the influence of self-concept versus feedback on learning, averaged at .253 (SD = .142). This value suggests that while external feedback predominantly guides participants’ updates to trait utilities, the integration of their self-concept remains a notable component of the learning process.

I also thought that the authors could have provided more details on the fine-granularity approach. I had some understanding of this idea from having read Frolichs et al. (2022) before, but I had to go back and review that paper to grasp the models in the present work.

Thanks for raising this issue here. We have now included a clear example in our manuscript to illustrate how prediction errors are spread to related traits [lines 645-652]:

*For instance, if a participant experiences a prediction error (PE) of '30' for the trait 'Responsible,' this PE will influence the updates of related traits in subsequent evaluations. Suppose 'Responsible' correlates with 'Punctual' .5. The update to 'Punctual' would then involve half of the prediction error received for the trait 'Responsible' calculated as $PE_{Responsible} * .5 [r(Responsible, Punctual)]$. This adjustment is further shaped by the learning rate, a (free) parameter that quantifies participants' responsiveness to Pes.*

Finally, two additional analyses questions: 1) First, Model 4 included a “free parameter gamma [γ (bounded between 0 and 1)] as a balancing factor to weigh the contribution of self-ratings against the learning-based predictions for each trial.” On average, was gamma skewed towards self-ratings or the learning-based predictions? Did individual differences in gamma correlated with self-concept clarity or self-esteem? 2) Second, since the authors used positive and negative trait terms (and potentially, people may display asymmetries in how they learn about positive and negative information that’s self-relevant), did the authors consider accounting for trait valence in their models? (For example, modeling separate learning rates for positive and negative traits?).

We thank the reviewer for their comments and suggestions. As mentioned in our previous response, the value of gamma was skewed toward feedback-informed learning. This information has been added as a part of the expansion of the results section you requested above. We have also included correlations between gamma and SCC and SE in the results section [lines 766-770]: *Finally, we aimed to test whether our computational parameters α and γ were correlated with measures of self-concept clarity and self-esteem. We found a significant and positive correlation between γ and SCC ($r(81) = .345, p = .001$) and a marginally significant and positive correlation between γ and SE ($r(81) = .198, p = .07$). No correlations were found between the learning rate and SCC or SE.*

We also appreciate your suggestion regarding the potential impact of trait valence on learning dynamics. Indeed, we tested the possibility that individuals might respond differently to positive versus negative traits, exactly as you suggested (separate LRs). This test was conducted to ensure a comprehensive understanding of the learning dynamics, even though we did not have a strong initial hypothesis for this specific aspect. The results of this and other explorations, however, confirmed that none of the alternative models outperformed Model 4. We chose to focus the manuscript on our target models to maintain clarity and relevance to our research aims, but we acknowledge the importance of your point and are open to include this analysis in the manuscript if you believe it is needed.

8. Lastly, I noted a few wording issues that could addressed:

- The authors write, “Yet, research indicates that measures based on these indicators do not accurately predict global indicators of SCC, nor do they exhibit strong correlations among them.” The latter part of this sentence might read more clearly as “...nor are they strongly intercorrelated.”

Thank you for your observation, we made the suggested change.

- The authors write, “In real-life scenarios, this involves that individuals need to map socially shared perceptions of which behaviors are appropriate...” – the wording of “involves that” is a little confusing; is a word missing perhaps?

Thank you for noticing this issue, the previous sentence has been rephrased to *That is, they need to map socially shared perceptions of which behaviors are appropriate and effective for achieving available goals in the landscape of their social contexts.*

- The authors write, “These strategies would allow refining their environmental models 296 while controlling its potential impact on SCC.” What is “its” referring to here?

We thank the reviewer for this comment. This sentence has been removed from the manuscript.

- Finally (and this is less of a wording issue and more a clarity issue), the authors write that they “...demonstrated that when the learning process is motivated (i.e., reduce SUD), individuals' current self-representations play an important role in structuring the social learning process.” How is “motivated” being used here? Is the implication that there is a chronic need to reduce SUD? Or was there something in the procedures that explicitly motivated this?

We thank the reviewer for the query regarding the use of 'motivated' in our manuscript. Here, following the rationale of our proposal and results, 'motivated' refers to the psychological drive where individuals are predisposed to align their learning of trait utilities with their self-concept, thereby avoiding the maximization Self-Utility Distance (SUD). To enhance clarity, we have slightly modified the text to: 'Here, we demonstrated that when the learning process is *potentially motivated (i.e., by the need to reduce SUD)*'.

Reviewer #3 (Remarks to the Author):

Review of the paper Self-Utility Distance: A Computational Approach to Understanding Self-Concept Clarity.

The paper introduces the concept of Self-Utility Distance (SUD), which is the absolute difference between the perceived values of specific personal characteristics and their perceived utility. The authors relate SUD to Self-Concept Clarity and, especially in the first study, to Self Evaluation; the authors adopt a functional perspective, where they consider the function of the concept they introduce to the adaptation of the individual to the social environment. In doing so, they adopt the perspective of reinforcement learning, one of the dominant mechanisms in current AI. Two empirical studies are presented as the test of the theory. The first one uses correlational design and multiple regression to examine the relations between SUD, Self-Clarity, and Self-evaluation. The second one is devoted to mechanisms underlying the formation of judgments concerning the utility of traits and examines the role of feedback from peers and perceptions of own traits in estimating the utility of specific traits. After providing their own judgment, the participants were presented with their peers' mean judgments of the utility of traits. The design of the study and the data analysis strategy were similar to the one used by Frolichs (2021), who studied the effects of peers' feedback on the perception of traits of others. Five computational models were used to discover the learning strategy from social feedback.

The paper's theoretical approach is interesting and worth publishing. The analysis of how the Self-Structure regulates behavior to increase adaptation, considering the environmental utility

of specific traits, is especially interesting and novel, especially concerning the use of reinforcement learning as the mechanism underlying the formation of the Self-structure. Also, introducing the concept of the utility of self-traits is novel. The concept of the Self-Utility Distance may define a new, important characteristic of the Self-Structure. However, this concept is also close in meaning and measurement to Self-Discrepancy (Higgins 1987). Self-discrepancy is the difference between the actual Self and the ideal Self. The discrepancy can also be measured between the actual Self and the Ought self. The central question to both the proposed theory and empirical studies is how the utility of a trait is related to its ideal value, or the value expected by social norms. This requires both theoretical discussion and an empirical investigation. The authors should explicitly discuss the relation of the Self-Utility Distance to Self-discrepancy. It also might help to replicate the first study measuring Self-discrepancy to examine if the concept of Self-Utility Distance explains unique variance not explained by Self-discrepancy, then the proposed concept presents a unique contribution to the theory of Self. Otherwise, it may be a renaming of one of the primary concepts of the Self, Self-discrepancy. There is considerable literature in psychology that discusses the psychological consequences of Self-discrepancy (e.g. Barnett, Moore, , & Harp, 2017; Higgins, Bond, Klein, R., & Strauman,1986).. Acknowledging this literature would increase the scholarship of the paper. The relationship of the proposed concept to similar existing concepts should be described in more detail.

We thank the reviewer for their thoughtful evaluation of our manuscript and for recognizing the potential of the Self-Utility Distance (SUD) concept to contribute significantly to the understanding of self-concept clarity. We are grateful for the positive feedback on the innovative use of reinforcement learning in our framework and for appreciating the potential of SUD.

In response to the reviewer's insightful comments on the theoretical positioning of Self-Utility Distance (SUD) in relation to Self-Discrepancy Theory (SDT), we have incorporated a new study (now Study 2 in the revised manuscript) comparing SUD with Ideal-Self Discrepancy (ISD) and Ought-Self Discrepancy (OSD), both conceptually and empirically. This study examines whether SUD explains unique variance in self-concept clarity (SCC) and self-esteem (SE) beyond ISD and OSD. Our findings confirm that SUD provides incremental validity in predicting SCC and potentially SE, highlighting its unique utility-based contribution to self-evaluation. This revision underscores that SUD is not merely a redefinition of self-discrepancy but introduces a novel theoretical construct with distinct implications. Additionally, for clarity and coherence, the study based on computational models has been relabelled as Study 3.

The concept of utility needs to be better defined because it is a central concept in the proposed model. The similarity of the Self-Utility Distance to existing concepts could also be better judged if the paper in the method section described the exact wording of the instructions to rate the trait utility in more detail. It is crucial because it is unclear if the participants were asked to estimate the importance of each trait for success or what value of this trait is optimal for success.

We thank the reviewer for highlighting the need to better define the concept of utility and clarify how participants evaluated it.

In our framework, utility refers to the perceived functional value of traits in contributing to achieving success or avoiding negative outcomes in a given context. This aligns with the

concept of expected cumulative rewards in reinforcement learning and reflects the subjective estimation of a trait's contribution to adaptation and goal achievement within individuals' environment. As such, our current definition aligns more with the notion of *importance of each trait for success* your proposed. The delineation and clarity of our operational definition of utility also benefited from the addition of the new study, in which we included additional information. For example [lines 372-377]:

The defining strength of SUD lies in its foundation on utility—a concept inherently computational that involves a subjective estimation of the capacity of self-attributes to maximize rewards or avoid harm in individuals' current life settings. That is, it quantifies their capacity to promote adaptation according to the perceived reward structure of the environment. This computational definition allows to conceptualize SUDs much like unresolved prediction errors in reinforcement learning.

and [lines 511-520]:

SUD emphasizes the functional mismatch between self-perceptions and their perceived utility in individuals' current life circumstances. In reinforcement learning, utility is a quantifiable measure of expected cumulative rewards associated with specific states, actions, or decisions. By framing SUD as the discrepancy between self-perceptions and their functional utility, we provide a construct that aligns with the adaptive mechanisms underlying learning processes. As such, SUD measures individuals' perceived "necessary adaptive changes" tied to current self-evaluations, which, akin to modifying behavioral strategies in RL paradigms, might trigger re-evaluation of the current self-structure to match the perceived functional value of self-attributes.

Following the reviewer's suggestion, we have also revised the methods section to include more details about the instructions provided to the participants [lines 227-235]:

Note that, before providing their estimations, participants were introduced to a definition of utility. We instructed them to consider utility as the capacity of each trait to provide them with positive consequences or help them avoid negative consequences in their current life settings. We also instructed them to consider the utility of each trait 'in general', together with a brief example ["For instance, if you encounter the trait 'Ambitious', you need to evaluate whether expressing this trait has the capacity to lead to positive outcomes or generate negative consequences in your life, in general, and as it is right now."].

In sum, the paper is worth publishing if the proposed SUD concept differs from the Self-Discrepancy concept. Answering the question of whether it is different requires a more explicit definition of the utility of traits and a description of rating institutions. An additional study might also be required to examine whether the two concepts can be empirically distinguished.

We hope that the inclusion of the new study effectively addresses your concerns and better clarifies the novel contributions of our work. We appreciate the opportunity to refine our manuscript and thank you for guiding us to strengthen the theoretical foundation and empirical validation of SUD.

Reviewer #1 (Remarks to the Author):

The authors have done an excellent job addressing reviewer comments. In particular, the revision is thorough in clarifying the methodological and theoretical approach, along with adding new data and analyses to further clarify the relationships.

We thank the reviewer for their constructive feedback. We are pleased that our revisions have clarified both the methodological and theoretical aspects of the work, and that the new data and analyses further illuminate the relationships we studied. Their comments have been very helpful in strengthening the manuscript.

I do have one minor suggestion for the authors. In the response letter, they note that they conducted additional analyses in which initial trait utilities in Models 2-3 were based on participants' self-ratings. In these models, the results held constant. I think this is an important finding: it supports the idea that initial ratings have an ongoing and consistent effect--potentially a motivational one--despite new learning, above and beyond setting a starting point for learning. These analyses seem worth mentioning, at least in the supplemental materials. Other than this suggestion, I think the manuscript can be accepted as is.

We thank the reviewer for this suggestion. We agree that this finding is important and have now incorporated a reference to these additional analyses in the main text [Lines 767-771]

We additionally performed analyses in which Models 2 and 3 were initialized with participants' self-ratings, and found that the results remained consistent (see Supplementary Materials, Section S3), reinforcing the notion that the effect of individuals' current self-concept parametrized in Model 4 exerts a persistent, potentially motivational influence on the learning process.

Along with a brief description of the results in the Supplementary Materials.

To test whether initial self-ratings merely serve as a starting point or exert a sustained, motivational influence on the learning process, we re-estimated Models 2 and 3 using each participant's own self-ratings as the initial trait utility values. Using the same Hierarchical Bayesian Inference (HBI) procedure, we found that the model comparisons remained robust, with Model 4 still emerging as the winning model (model frequency = 91.5%, Protected Exceedance Probability = 1). This result suggests that the initial self-ratings persistently influence the updating process, reinforcing the interpretation that individuals are motivated to align new feedback with their established self-concept.

Reviewer #2 (Remarks to the Author):

We thank the reviewer for the thorough and thoughtful comments. The time and effort invested in reviewing the manuscript are greatly appreciated, and the insights provided have been invaluable

The authors have thoughtfully responded to the comments I provided in my initial review. I appreciate their careful revision of this manuscript. While I have a few minor thoughts remaining, in my view, the manuscript is greatly improved. Thank you very much for the

opportunity to consider this work!

1. I think this is ultimately a small issue but upon reading the new Study 2 (and rereading Study 1), I think I struggled at times to identify the precise placement/role of certain variables—especially self-esteem—in a logical progression from one to the next. For example, at different times, self-esteem is treated as a competing predictor, a potential confound to be controlled, and an outcome variable. That said, I recognize that maybe my thinking here is a bit reductive and that it's premature to assume a particular serial (or parallel) structure here. Indeed, as the authors state on pg. 9, "...the directionality of the relationship between SCC and SE has not been robustly established. Including SUD in longitudinal studies could clarify the relationship between SCC and SE 19–22, while also allowing to assess whether SCC and SE reciprocally influence SUD over time."

I think ultimately, it would just help to have a line or two motivating the analysis in the new Study 2 that casts SE as the outcome (i.e., "Moreover, we explored whether SUD could also show incremental validity over SUD in the prediction of Self Esteem.") Also, I think that second SUD should be "the SDT measures" or something like that.

We value the reviewer's comment in this regard, and agree that some clarification is needed. We pursued this analysis because while Self-Utility Distance (SUD) primarily captures structural misalignments it may also carry direct affective consequences. In other words, if unresolved tension between one's self-attributes and their perceived functional utility disrupt the coherence of self-representations, they may concurrently disrupt self-esteem. By testing SE as an outcome, we aimed to determine whether SUD can explain unique variance in self-esteem above and beyond that accounted for by the traditional Self-Discrepancy Theory measures (i.e., ideal-self and ought-self discrepancies). In line with the reviewer's suggestion, we added a brief sentence in the manuscript for clarification [Lines 467-469]:

To explore whether SUD can be understood as an affective signal similar to the components of the SDT we aimed to reproduce the same analyses but focusing on Self-Esteem as the dependent variable.

We also thank the reviewer for noticing the error in the sentence "second SUD should be "the SDT measures". We made the correction as suggested.

2. The authors link SUD to the concept of prediction error several times – i.e., "This computational definition allows to conceptualize SUDs much like unresolved prediction errors in reinforcement learning," "...similar to prediction errors, SUDs may also be aversive to the individual, triggering negative emotional responses that may affect SE," etc. This certainly seems reasonable. Recent work has begun to distinguish between reward (or outcome) prediction errors and *affective* (or emotion) prediction errors (Vollberg & Sander, 2024; Vollberg, & Cikara, 2024; Heffner, Frömer, Nassar, & FeldmanHall, 2024). Can the authors speculate whether the discrepancies captured by SUD map more onto one type of PE vs. the other?

We thank the reviewer for this interesting suggestion. The particularity of SUD is that it does not necessarily involve either reward or affective PEs, since subjective utility estimations computed by the individual could, in principle, be accurate. However, we believe that situating

SUD into this context might be valuable and help constructing mutually informative insights among research lines. We opted to avoid making inferences about their specific relationship but added a brief paragraph in our manuscript to motivate future research [Lines 933-940]

To further elucidate the nature and functioning of Self-Utility Distance (SUD), it appears beneficial to explore its relationship with well-established error-like signals, such as reward and affective prediction errors (PEs)^{95,96}. We defined SUD as an error signal that indicates a necessary adjustment that has not been undertaken by the individual, due to the inherent stability in behavior and self-concept representations. Future research should investigate whether this error signal or its potential disruptive impact are independent of whether anticipated rewards or emotional states are accurately estimated by the individual.

3. In their response letter, the authors referred to several other analyses that they conducted in response to my comments, which they chose not to include in the manuscript (i.e., regarding trait valence and participant age). I agree with their assessments that the analyses are not critical to this work.

We thank the reviewer's feedback. We are happy with having further analyzed our data and assumptions.

Regarding the racial breakdown of their sample, the authors indicated that race/ethnicity is not typically collected, as "over 90% of graduate students in Barcelona are white, which further reflects the homogeneity in the demographic composition of our sample." This makes sense, but I'd recommend simply stating that in text.

We followed the reviewer's suggestion and now included this information in the text [Lines 217-218]:

Given the small variability in terms of race and ethnicity in participants enrolling from the university's lab panel, race/ethnicity data was not collected.

That said, the new Study 2 was conducted on Prolific, which does typically yield more racially diverse samples. Did the authors still choose not to collect this from their participants? Moreover, what geographic / nationality / language filters were used when conducting this data collection?

We thank the reviewer for this last remark on sociodemographic data. For Study 2, conducted on Prolific, we did not actively collect race/ethnicity data; however, based on the (limited) demographic information provided through Prolific by default, approximately 80% of our participants self-identified as white, with limited variability observed. We specifically recruited Spanish speakers without imposing any geographic restrictions, and age was restricted to fall within a range of 18–40 years. We have now noted this in our manuscript [Lines 412-415]:

For this study we recruited Spanish-speaking participants with an age range of 18–40 years without imposing any geographic restrictions. Although we did not actively collect race/ethnicity data, demographic information provided through Prolific indicated that approximately 80% of the sample self-identified as white

4. Finally, a few typographical errors that I noted:

- pg. 2: "...comprising various personal traits such as 'Sociable' or 'Anxious'" – I think "personal" should be "personality"

We agree that "personality traits" is a more conventional term in this context. We have updated our wording.

- pg. 9: "The actual self includes traits that an individual believes to possess." – should that sentence end with "...believes that they possess"?

We agree with the reviewer and made the suggested change.

- pg. 9: In "The theory suggests that these self-discrepancies have wide variety of impacts on individuals' emotional outcomes..." it looks like "a" is missing between "have" and "wide"
- On pg. 10, "Ought-Self" is misspelled as "Ough-Self"

We thank the reviewer for catching that misspelling. We have now corrected the sentence.